# Neural Arithmetic Units

**Andreas Madsen**
Computationally Demanding
amwebdk@gmail.com

**Alexander Rosenberg Johansen**
Technical University of Denmark
aler@dtu.dk

## Abstract

Neural networks can approximate complex functions, but they struggle to perform exact arithmetic operations over real numbers. The lack of inductive bias for arithmetic operations leaves neural networks without the underlying logic necessary to extrapolate on tasks such as addition, subtraction, and multiplication. We present two new neural network components: the Neural Addition Unit (NAU), which can learn exact addition and subtraction; and the Neural Multiplication Unit (NMU) that can multiply subsets of a vector. The NMU is, to our knowledge, the first arithmetic neural network component that can learn to multiply elements from a vector, when the hidden size is large. The two new components draw inspiration from a theoretical analysis of recently proposed arithmetic components. We find that careful initialization, restricting parameter space, and regularizing for sparsity is important when optimizing the NAU and NMU. Our proposed units NAU and NMU, compared with previous neural units, converge more consistently, have fewer parameters, learn faster, can converge for larger hidden sizes, obtain sparse and meaningful weights, and can extrapolate to negative and small values.[1]

## 1 Introduction

When studying intelligence, insects, reptiles, and humans have been found to possess neurons with the capacity to hold integers, real numbers, and perform arithmetic operations (Nieder, 2016; Rugani et al., 2009; Gallistel, 2018). In our quest to mimic intelligence, we have put much faith in neural networks, which in turn has provided unparalleled and often superhuman performance in tasks requiring high cognitive abilities (Silver et al., 2016; Devlin et al., 2018; OpenAI et al., 2018). However, when using neural networks to solve simple arithmetic problems, such as counting, multiplication, or comparison, they systematically fail to extrapolate onto unseen ranges (Lake & Baroni, 2018; Suzgun et al., 2019; Trask et al., 2018). The absence of inductive bias makes it difficult for neural networks to extrapolate well on arithmetic tasks as they lack the underlying logic to represent the required operations.

A neural component that can solve arithmetic problems should be able to: take an arbitrary hidden input, learn to select the appropriate elements, and apply the desired arithmetic operation. A recent attempt to achieve this goal is the Neural Arithmetic Logic Unit (NALU) by Trask et al. (2018).

The NALU models the inductive bias explicitly via two sub-units: the $NAC_+$ for addition/subtraction and the $NAC_\bullet$ for multiplication/division. The sub-units are softly gated between, using a sigmoid function, to exclusively select one of the sub-units. However, we find that the soft gating-mechanism and the $NAC_\bullet$ are fragile and hard to learn.

In this paper, we analyze and improve upon the $NAC_+$ and $NAC_\bullet$ with respect to addition, subtraction, and multiplication. Our proposed improvements, namely the Neural Addition Unit (NAU) and Neural Multiplication Unit (NMU), are more theoretically founded and improve performance regarding stability, speed of convergence, and interpretability of weights. Most importantly, the NMU supports both negative and small numbers and a large hidden input-size, which is paramount as neural networks are overparameterized and hidden values are often unbounded.

The improvements, which are based on a theoretical analysis of the NALU and its components, are achieved by a simplification of the parameter matrix for a better gradient signal, a sparsity regularizer, and a new multiplication unit that can be optimally initialized. The NMU does not support division.

---

[1]Implementation is available on GitHub: https://github.com/AndreasMadsen/stable-nalu.

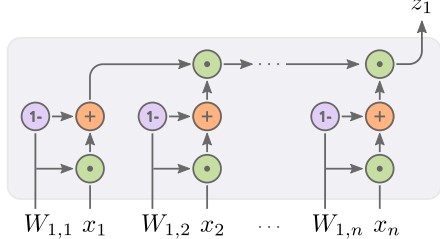

Figure 1: Visualization of the NMU, where the weights ($W_{i,j}$) controls gating between 1 (identity) or $x_i$, each intermediate result is then multiplied explicitly to form $z_j$.

However, we find that the $NAC_\bullet$ in practice also only supports multiplication and cannot learn division (theoretical analysis on division discussed in section 2.3).

To analyze the impact of each improvement, we introduce several variants of the $NAC_\bullet$. We find that allowing division makes optimization for multiplication harder, linear and regularized weights improve convergence, and the NMU way of multiplying is critical when increasing the hidden size.

Furthermore, we improve upon existing benchmarks in Trask et al. (2018) by expanding the "simple function task", expanding "MNIST Counting and Arithmetic Tasks" with a multiplicative task, and using an improved success-criterion Madsen & Johansen (2019). This success-criterion is important because the arithmetic units are solving a logical problem. We propose the MNIST multiplication variant as we want to test the NMU's and $NAC_\bullet$'s ability to learn from real data and extrapolate.

## 1.1 LEARNING A 10 PARAMETER FUNCTION

Consider the static function $t = (x_1 + x_2) \cdot (x_1 + x_2 + x_3 + x_4)$ for $x \in \mathbb{R}^4$. To illustrate the ability of $NAC_\bullet$, NALU, and our proposed NMU, we conduct 100 experiments for each model to learn this function. Table 1 shows that the NMU has a higher success rate and converges faster.

Table 1: Comparison of the success-rate, when the model converged, and the sparsity error for all weight matrices, with 95% confidence interval on the $t = (x_1 + x_2) \cdot (x_1 + x_2 + x_3 + x_4)$ task. Each value is a summary of 100 different seeds.

| Op | Model | Success | Solved at iteration step | | Sparsity error |
|----|-------|---------|--------------------------|------|----------------|
| | | Rate | Median | Mean | Mean |
| | $NAC_\bullet$ | $13\%\,^{+8\%}_{-5\%}$ | $5.5 \cdot 10^4$ | $5.9 \cdot 10^4\,^{+7.8 \cdot 10^3}_{-6.6 \cdot 10^3}$ | $7.5 \cdot 10^{-6}\,^{+2.0 \cdot 10^{-6}}_{-2.0 \cdot 10^{-6}}$ |
| $\times$ | NALU | $26\%\,^{+9\%}_{-8\%}$ | $7.0 \cdot 10^4$ | $7.8 \cdot 10^4\,^{+6.2 \cdot 10^3}_{-8.6 \cdot 10^3}$ | $9.2 \cdot 10^{-6}\,^{+1.7 \cdot 10^{-6}}_{-1.7 \cdot 10^{-6}}$ |
| | NMU | $\mathbf{94\%}\,^{+3\%}_{-6\%}$ | $\mathbf{1.4 \cdot 10^4}$ | $\mathbf{1.4 \cdot 10^4}\,^{+2.2 \cdot 10^2}_{-2.1 \cdot 10^2}$ | $\mathbf{2.6 \cdot 10^{-8}}\,^{+6.4 \cdot 10^{-9}}_{-6.4 \cdot 10^{-9}}$ |

## 2 INTRODUCING DIFFERENTIABLE BINARY ARITHMETIC OPERATIONS

We define our problem as learning a set of static arithmetic operations between selected elements of a vector. E.g. for a vector $\mathbf{x}$ learn the function $(x_5 + x_1) \cdot x_7$. The approach taking in this paper is to develop a unit for addition/subtraction and a unit for multiplication, and then let each unit decide which inputs to include using backpropagation.

We develop these units by taking inspiration from a theoretical analysis of Neural Arithmetic Logic Unit (NALU) by Trask et al. (2018).

## 2.1 INTRODUCING NALU

The Neural Arithmetic Logic Unit (NALU) consists of two sub-units; the $NAC_+$ and $NAC_\bullet$. The sub-units represent either the $\{+, -\}$ or the $\{\times, \div\}$ operations. The NALU then assumes that either $NAC_+$ or $NAC_\bullet$ will be selected exclusively, using a sigmoid gating-mechanism.

The NAC$_+$ and NAC$_\bullet$ are defined accordingly,

$$W_{h_\ell, h_{\ell-1}} = \tanh(\hat{W}_{h_\ell, h_{\ell-1}})\sigma(\hat{M}_{h_\ell, h_{\ell-1}}) \tag{1}$$

$$\text{NAC}_+ : z_{h_\ell} = \sum_{h_{\ell-1}=1}^{H_{\ell-1}} W_{h_\ell, h_{\ell-1}} z_{h_{\ell-1}} \tag{2}$$

$$\text{NAC}_\bullet : z_{h_\ell} = \exp\left(\sum_{h_{\ell-1}=1}^{H_{\ell-1}} W_{h_\ell, h_{\ell-1}} \log(|z_{h_{\ell-1}}| + \epsilon)\right) \tag{3}$$

where $\hat{\mathbf{W}}, \hat{\mathbf{M}} \in \mathbb{R}^{H_\ell \times H_{\ell-1}}$ are weight matrices and $z_{h_{\ell-1}}$ is the input. The matrices are combined using a tanh-sigmoid transformation to bias the parameters towards a $\{-1, 0, 1\}$ solution. Having $\{-1, 0, 1\}$ allows NAC$_+$ to compute exact $\{+, -\}$ operations between elements of a vector. The NAC$_\bullet$ uses an exponential-log transformation for the $\{\times, \div\}$ operations, which works within $\epsilon$ precision and for positive inputs only.

The NALU combines these units with a gating mechanism $\mathbf{z} = \mathbf{g} \odot \text{NAC}_+ + (1 - \mathbf{g}) \odot \text{NAC}_\bullet$ given $\mathbf{g} = \sigma(\mathbf{Gx})$. Thus allowing NALU to decide between all of $\{+, -, \times, \div\}$ using backpropagation.

## 2.2 Weight matrix construction and the Neural Addition Unit

Glorot & Bengio (2010) show that $E[z_{h_\ell}] = 0$ at initialization is a desired property, as it prevents an explosion of both the output and the gradients. To satisfy this property with $W_{h_{\ell-1}, h_\ell} = \tanh(\hat{W}_{h_{\ell-1}, h_\ell})\sigma(\hat{M}_{h_{\ell-1}, h_\ell})$, an initialization must satisfy $E[\tanh(\hat{W}_{h_{\ell-1}, h_\ell})] = 0$. In NALU, this initialization is unbiased as it samples evenly between $+$ and $-$, or $\times$ and $\div$. Unfortunately, this initialization also causes the expectation of the gradient to become zero, as shown in (4).

$$E\left[\frac{\partial \mathcal{L}}{\partial \hat{M}_{h_{\ell-1}, h_\ell}}\right] = E\left[\frac{\partial \mathcal{L}}{\partial W_{h_{\ell-1}, h_\ell}}\right] E\left[\tanh(\hat{W}_{h_{\ell-1}, h_\ell})\right] E\left[\sigma'(\hat{M}_{h_{\ell-1}, h_\ell})\right] = 0 \tag{4}$$

Besides the issue of initialization, our empirical analysis (table 2) shows that this weight construction (1) do not create the desired bias for $\{-1, 0, 1\}$. This bias is desired as it restricts the solution space to exact addition, and in section 2.5 also exact multiplication, which is an intrinsic property of an underlying arithmetic function. However, this bias does not necessarily restrict the output space as a plain linear transformation will always be able to scale values accordingly.

To solve these issues, we add a sparsifying regularizer to the loss function ($\mathcal{L} = \hat{\mathcal{L}} + \lambda_{\text{sparse}}\mathcal{R}_{\ell,\text{sparse}}$) and use a simple linear construction, where $W_{h_{\ell-1}, h_\ell}$ is clamped to $[-1, 1]$ in each iteration.

$$W_{h_{\ell-1}, h_\ell} = \min(\max(W_{h_{\ell-1}, h_\ell}, -1), 1), \tag{5}$$

$$\mathcal{R}_{\ell,\text{sparse}} = \frac{1}{H_\ell \cdot H_{\ell-1}} \sum_{h_\ell=1}^{H_\ell} \sum_{h_{\ell-1}=1}^{H_{\ell-1}} \min\left(|W_{h_{\ell-1}, h_\ell}|, 1 - |W_{h_{\ell-1}, h_\ell}|\right) \tag{6}$$

$$\text{NAU} : z_{h_\ell} = \sum_{h_{\ell-1}=1}^{H_{\ell-1}} W_{h_\ell, h_{\ell-1}} z_{h_{\ell-1}} \tag{7}$$

## 2.3 Challenges of division

The NAC$_\bullet$, as formulated in equation 3, has the capability to compute exact multiplication and division, or more precisely multiplication of the inverse of elements from a vector, when a weight in $W_{h_{\ell-1}, h_\ell}$ is $-1$.

However, this flexibility creates critical optimization challenges. By expanding the exp-log-transformation, NAC$_\bullet$ can be expressed as

$$\text{NAC}_\bullet : \ z_{h_\ell} = \prod_{h_{\ell-1}=1}^{H_{\ell-1}} \left(|z_{h_{\ell-1}}| + \epsilon\right)^{W_{h_\ell, h_{\ell-1}}} . \tag{8}$$

In equation (8), if $|z_{h_{\ell-1}}|$ is near zero ($E[z_{h_{\ell-1}}] = 0$ is a desired property when initializing (Glorot & Bengio, 2010)), $W_{h_{\ell-1}, h_\ell}$ is negative, and $\epsilon$ is small, then the output will explode. This issue is present even for a reasonably large $\epsilon$ value (such as $\epsilon = 0.1$), and just a slightly negative $W_{h_{\ell-1}, h_\ell}$, as visualized in figure 2. Also note that the curvature can cause convergence to an unstable area.

This singularity issue in the optimization space also makes multiplication challenging, which further suggests that supporting division is undesirable. These observations are also found empirically in Trask et al. (2018, table 1) and Appendix C.7.

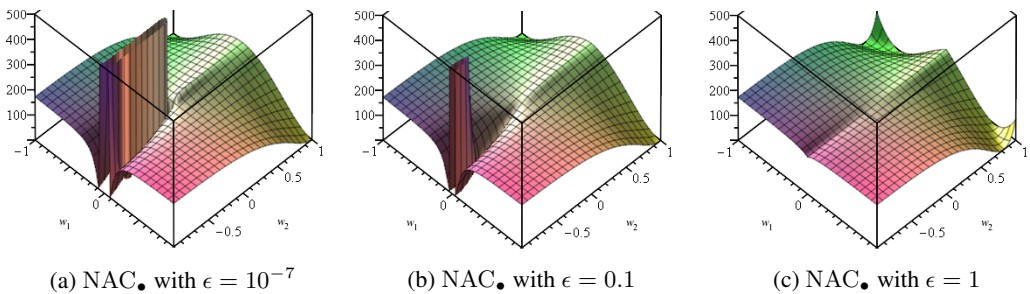

(a) NAC$_\bullet$ with $\epsilon = 10^{-7}$    (b) NAC$_\bullet$ with $\epsilon = 0.1$    (c) NAC$_\bullet$ with $\epsilon = 1$

Figure 2: RMS loss curvature for a NAC$_+$ unit followed by a NAC$_\bullet$. The weight matrices are constrained to $\mathbf{W}_1 = \left[ \begin{smallmatrix} w_1 & w_1 & 0 & 0 \\ w_1 & w_1 & w_1 & w_1 \end{smallmatrix} \right]$, $\mathbf{W}_2 = \left[ \begin{smallmatrix} w_2 & w_2 \end{smallmatrix} \right]$. The problem is $(x_1 + x_2) \cdot (x_1 + x_2 + x_3 + x_4)$ for $x = (1, 1.2, 1.8, 2)$. The solution is $w_1 = w_2 = 1$ in (a), with many unstable alternatives.

## 2.4 INITIALIZATION OF NAC$_\bullet$

Initialization is important for fast and consistent convergence. A desired property is that weights are initialized such that $E[z_{h_\ell}] = 0$ (Glorot & Bengio, 2010). Using second order Taylor approximation and assuming all $z_{h_{\ell-1}}$ are uncorrelated; the expectation of NAC$_\bullet$ can be estimated as

$$E[z_{h_\ell}] \approx \left(1 + \frac{1}{2} Var[W_{h_\ell, h_{\ell-1}}] \log(|E[z_{h_{\ell-1}}]| + \epsilon)^2\right)^{H_{\ell-1}} \Rightarrow E[z_{h_\ell}] > 1. \tag{9}$$

As shown in equation 9, satisfying $E[z_{h_\ell}] = 0$ for NAC$_\bullet$ is likely impossible. The variance cannot be input-independently initialized and is expected to explode (proofs in Appendix B.3).

## 2.5 THE NEURAL MULTIPLICATION UNIT

To solve the the gradient and initialization challenges for NAC$_\bullet$ we propose a new unit for multiplication: the Neural Multiplication Unit (NMU)

$$W_{h_{\ell-1}, h_\ell} = \min(\max(W_{h_{\ell-1}, h_\ell}, 0), 1), \tag{10}$$

$$\mathcal{R}_{\ell, \text{sparse}} = \frac{1}{H_\ell \cdot H_{\ell-1}} \sum_{h_\ell=1}^{H_\ell} \sum_{h_{\ell-1}=1}^{H_{\ell-1}} \min\left(W_{h_{\ell-1}, h_\ell}, 1 - W_{h_{\ell-1}, h_\ell}\right) \tag{11}$$

$$\text{NMU} : \ z_{h_\ell} = \prod_{h_{\ell-1}=1}^{H_{\ell-1}} \left(W_{h_{\ell-1}, h_\ell} z_{h_{\ell-1}} + 1 - W_{h_{\ell-1}, h_\ell}\right) \tag{12}$$

The NMU is regularized similar to the NAU and has a multiplicative identity when $W_{h_{\ell-1}, h_\ell} = 0$. The NMU does not support division by design. As opposed to the NAC$_\bullet$, the NMU can represent input of both negative and positive values and is not $\epsilon$ bounded, which allows the NMU to extrapolate to $z_{h_{\ell-1}}$ that are negative or smaller than $\epsilon$. Its gradients are derived in Appendix A.3.

### 2.6 MOMENTS AND INITIALIZATION

The NAU is a linear layer and can be initialized using Glorot & Bengio (2010). The $\text{NAC}_+$ unit can also achieve an ideal initialization, although it is less trivial (details in Appendix B.2).

The NMU is initialized with $E[W_{h_\ell,h_{\ell-1}}] = 1/2$. Assuming all $z_{h_{\ell-1}}$ are uncorrelated, and $E[z_{h_{\ell-1}}] = 0$, which is the case for most neural units (Glorot & Bengio, 2010), the expectation can be approximated to

$$E[z_{h_\ell}] \approx \left(\frac{1}{2}\right)^{H_{\ell-1}},\tag{13}$$

which approaches zero for $H_{\ell-1} \to \infty$ (see Appendix B.4). The NMU can, assuming $Var[z_{h_{\ell-1}}] = 1$ and $H_{\ell-1}$ is large, be optimally initialized with $Var[W_{h_{\ell-1},h_\ell}] = \frac{1}{4}$ (proof in Appendix B.4.3).

### 2.7 REGULARIZER SCALING

We use the regularizer scaling as defined in (14). We motivate this by observing optimization consists of two parts: a warmup period, where $W_{h_{\ell-1},h_\ell}$ should get close to the solution, unhindered by the sparsity regularizer, followed by a period where the solution is made sparse.

$$\lambda_{\text{sparse}} = \hat{\lambda}_{\text{sparse}} \max\left(\min\left(\frac{t - \lambda_{\text{start}}}{\lambda_{\text{end}} - \lambda_{\text{start}}}, 1\right), 0\right)\tag{14}$$

### 2.8 CHALLENGES OF GATING BETWEEN ADDITION AND MULTIPLICATION

The purpose of the gating-mechanism is to select either $\text{NAC}_+$ or $\text{NAC}_\bullet$ exclusively. This assumes that the correct sub-unit is selected by the NALU, since selecting the wrong sub-unit leaves no gradient signal for the correct sub-unit.

Empirically we find this assumption to be problematic. We observe that both sub-units converge at the beginning of training whereafter the gating-mechanism, seemingly random, converge towards either the addition or multiplication unit. Our study shows that gating behaves close to random for both NALU and a gated NMU/NAU variant. However, when the gate correctly selects multiplication our NMU converges much more consistently. We provide empirical analysis in Appendix C.5 for both NALU and a gated version of NAU/NMU.

As the output-size grows, randomly choosing the correct gating value becomes an exponential increasing problem. Because of these challenges we leave solving the issue of sparse gating for future work and focus on improving the sub-units $\text{NAC}_+$ and $\text{NAC}_\bullet$.

## 3 RELATED WORK

Pure neural models using convolutions, gating, differentiable memory, and/or attention architectures have attempted to learn arithmetic tasks through backpropagation (Kaiser & Sutskever, 2016; Kalchbrenner et al., 2016; Graves et al., 2014; Freivalds & Liepins, 2017). Some of these results have close to perfect extrapolation. However, the models are constrained to only work with well-defined arithmetic setups having no input redundancy, a single operation, and one-hot representations of numbers for input and output. Our proposed models does not have these restrictions.

The Neural Arithmetic Expression Calculator (Chen et al., 2018) can learn real number arithmetic by having neural network sub-components and repeatedly combine them through a memory-encoder-decoder architecture learned with hierarchical reinforcement learning. While this model has the ability to dynamically handle a larger variety of expressions compared to our solution they require an explicit definition of the operations, which we do not.

In our experiments, the NAU is used to do a subset-selection, which is then followed by either a summation or a multiplication. An alternative, fully differentiable version, is to use a gumbel-softmax that can perform exact subset-selection (Xie & Ermon, 2019). However, this is restricted to a predefined subset size, which is a strong assumption that our units are not limited by.

## 4 EXPERIMENTAL RESULTS

### 4.1 ARITHMETIC DATASETS

The arithmetic dataset is a replica of the "simple function task" by Trask et al. (2018). The goal is to sum two random contiguous subsets of a vector and apply an arithmetic operation as defined in (15)

$$t = \sum_{i=s_{1,\text{start}}}^{s_{1,\text{end}}} x_i \circ \sum_{i=s_{2,\text{start}}}^{s_{2,\text{end}}} x_i \quad \text{where } \mathbf{x} \in \mathbb{R}^n, x_i \sim \text{Uniform}[r_{\text{lower}}, r_{\text{upper}}], \circ \in \{+, -, \times\} \quad (15)$$

where $n$ (default 100), $U[r_{\text{lower}}, r_{\text{upper}}]$ (interpolation default is $U[1, 2]$ and extrapolation default is $U[2, 6]$), and other dataset parameters are used to assess learning capability (see details in Appendix C.1 and the effect of varying the parameters in Appendix C.4).

#### 4.1.1 MODEL EVALUATION

We define the success-criterion as a solution that is acceptably close to a perfect solution. To evaluate if a model instance solves the task consistently, we compare the MSE to a nearly-perfect solution on the extrapolation range over many seeds. If $\mathbf{W}_1, \mathbf{W}_2$ defines the weights of the fitted model, $\mathbf{W}_1^\epsilon$ is nearly-perfect, and $\mathbf{W}_2^*$ is perfect (example in equation 16), then the criteria for successful convergence is $\mathcal{L}_{\mathbf{W}_1, \mathbf{W}_2} < \mathcal{L}_{\mathbf{W}_1^\epsilon, \mathbf{W}_2^*}$, measured on the extrapolation error, for $\epsilon = 10^{-5}$. We report a 95% confidence interval using a binomial distribution (Wilson, 1927).

$$\mathbf{W}_1^\epsilon = \begin{bmatrix} 1 - \epsilon & 1 - \epsilon & 0 + \epsilon & 0 + \epsilon \\ 1 - \epsilon & 1 - \epsilon & 1 - \epsilon & 1 - \epsilon \end{bmatrix}, \mathbf{W}_2^* = \begin{bmatrix} 1 & 1 \end{bmatrix} \quad (16)$$

To measure the speed of convergence, we report the first iteration for which $\mathcal{L}_{\mathbf{W}_1, \mathbf{W}_2} < \mathcal{L}_{\mathbf{W}_1^\epsilon, \mathbf{W}_2^*}$ is satisfied, with a 95% confidence interval calculated using a gamma distribution with maximum likelihood profiling. Only instances that solved the task are included.

We assume an approximate discrete solution with parameters close to $\{-1, 0, 1\}$ is important for inferring exact arithmetic operations. To measure the sparsity, we introduce a sparsity error (defined in equation 17). Similar to the convergence metric, we only include model instances that did solve the task and report the 95% confidence interval, which is calculated using a beta distribution with maximum likelihood profiling.

$$E_{\text{sparsity}} = \max_{h_{\ell-1}, h_\ell} \min(|W_{h_{\ell-1}, h_\ell}|, |1 - |W_{h_{\ell-1}, h_\ell}||) \quad (17)$$

#### 4.1.2 ARITHMETIC OPERATION COMPARISON

We compare models on different arithmetic operations $\circ \in \{+, -, \times\}$. The multiplication models, NMU and NAC$_\bullet$, have an addition unit first, either NAU or NAC$_+$, followed by a multiplication unit. The addition/substraction models are two layers of the same unit. The NALU model consists of two NALU layers. See explicit definitions and regularization values in Appendix C.2.

Each experiment is trained for $5 \cdot 10^6$ iterations with early stopping by using the validation dataset, which is based on the interpolation range (details in Appendix C.2). The results are presented in table 2. For multiplication, the NMU succeeds more often and converges faster than the NAC$_\bullet$ and NALU. For addition and substraction, the NAU and NAC$_+$ has similar success-rate (100%), but the NAU is significantly faster at solving both of the the task. Moreover, the NAU reaches a significantly sparser solution than the NAC$_+$. Interestingly, a linear model has a hard time solving subtraction. A more extensive comparison is included in Appendix C.7 and an ablation study is included in Appendix C.3.

#### 4.1.3 EVALUATING THEORETICAL CLAIMS

To validate our theoretical claim, that the NMU model works better than NAC$_\bullet$ for a larger hidden input-size, we increase the hidden size of the network thereby adding redundant units. Redundant units are very common in neural networks, which are often overparameterized.

Additionally, the NMU model is, unlike the NAC$_\bullet$ model, capable of supporting inputs that are both negative and positive. To validate this empirically, the training and validation datasets are sampled for $U[-2, 2]$, and then tested on $U[-6, -2] \cup U[2, 6]$. The other ranges are defined in Appendix C.4.

Table 2: Comparison of: success-rate, first iteration reaching success, and sparsity error, all with 95% confidence interval on the "arithmetic datasets" task. Each value is a summary of 100 different seeds.

| Op | Model | Success | Solved at iteration step | | Sparsity error |
|---|---|---|---|---|---|
| | | Rate | Median | Mean | Mean |
| $\times$ | $\text{NAC}_\bullet$ | $31\% \, ^{+10\%}_{-8\%}$ | $2.8 \cdot 10^6$ | $3.0 \cdot 10^6 \, ^{+2.9 \cdot 10^5}_{-2.4 \cdot 10^5}$ | $5.8 \cdot 10^{-4} \, ^{+4.8 \cdot 10^{-4}}_{-2.6 \cdot 10^{-4}}$ |
| | NALU | $0\% \, ^{+4\%}_{-0\%}$ | — | — | — |
| | NMU | $\mathbf{98\%} \, ^{+1\%}_{-5\%}$ | $\mathbf{1.4 \cdot 10^6}$ | $\mathbf{1.5 \cdot 10^6} \, ^{+5.0 \cdot 10^4}_{-6.6 \cdot 10^4}$ | $\mathbf{4.2 \cdot 10^{-7}} \, ^{+2.9 \cdot 10^{-8}}_{-2.9 \cdot 10^{-8}}$ |
| $+$ | $\text{NAC}_+$ | $\mathbf{100\%} \, ^{+0\%}_{-4\%}$ | $2.5 \cdot 10^5$ | $4.9 \cdot 10^5 \, ^{+5.2 \cdot 10^4}_{-4.5 \cdot 10^4}$ | $2.3 \cdot 10^{-1} \, ^{+6.5 \cdot 10^{-3}}_{-6.5 \cdot 10^{-3}}$ |
| | Linear | $\mathbf{100\%} \, ^{+0\%}_{-4\%}$ | $6.1 \cdot 10^4$ | $\mathbf{6.3 \cdot 10^4} \, ^{+2.5 \cdot 10^3}_{-3.3 \cdot 10^3}$ | $2.5 \cdot 10^{-1} \, ^{+3.6 \cdot 10^{-4}}_{-3.6 \cdot 10^{-4}}$ |
| | NALU | $14\% \, ^{+8\%}_{-5\%}$ | $1.5 \cdot 10^6$ | $1.6 \cdot 10^6 \, ^{+3.8 \cdot 10^5}_{-3.3 \cdot 10^5}$ | $1.7 \cdot 10^{-1} \, ^{+2.7 \cdot 10^{-2}}_{-2.5 \cdot 10^{-2}}$ |
| | NAU | $\mathbf{100\%} \, ^{+0\%}_{-4\%}$ | $\mathbf{1.8 \cdot 10^4}$ | $3.9 \cdot 10^5 \, ^{+4.5 \cdot 10^4}_{-3.7 \cdot 10^4}$ | $\mathbf{3.2 \cdot 10^{-5}} \, ^{+1.3 \cdot 10^{-5}}_{-1.3 \cdot 10^{-5}}$ |
| $-$ | $\text{NAC}_+$ | $\mathbf{100\%} \, ^{+0\%}_{-4\%}$ | $9.0 \cdot 10^3$ | $3.7 \cdot 10^5 \, ^{+3.8 \cdot 10^4}_{-3.8 \cdot 10^4}$ | $2.3 \cdot 10^{-1} \, ^{+5.4 \cdot 10^{-3}}_{-5.4 \cdot 10^{-3}}$ |
| | Linear | $7\% \, ^{+7\%}_{-4\%}$ | $3.3 \cdot 10^6$ | $1.4 \cdot 10^6 \, ^{+7.0 \cdot 10^5}_{-6.1 \cdot 10^5}$ | $1.8 \cdot 10^{-1} \, ^{+7.2 \cdot 10^{-2}}_{-5.8 \cdot 10^{-2}}$ |
| | NALU | $14\% \, ^{+8\%}_{-5\%}$ | $1.9 \cdot 10^6$ | $1.9 \cdot 10^6 \, ^{+4.4 \cdot 10^5}_{-4.5 \cdot 10^5}$ | $2.1 \cdot 10^{-1} \, ^{+2.2 \cdot 10^{-2}}_{-2.2 \cdot 10^{-2}}$ |
| | NAU | $\mathbf{100\%} \, ^{+0\%}_{-4\%}$ | $\mathbf{5.0 \cdot 10^3}$ | $\mathbf{1.6 \cdot 10^5} \, ^{+1.7 \cdot 10^4}_{-1.6 \cdot 10^4}$ | $6.6 \cdot 10^{-2} \, ^{+2.5 \cdot 10^{-2}}_{-1.9 \cdot 10^{-2}}$ |

Finally, for a fair comparison we introduce two new units: A variant of $\text{NAC}_\bullet$, denoted $\text{NAC}_{\bullet,\sigma}$, that only supports multiplication by constraining the weights with $W = \sigma(\hat{W})$. And a variant, named $\text{NAC}_{\bullet,\text{NMU}}$, that uses clamped linear weights and sparsity regularization identically to the NMU.

Figure 3 shows that the NMU can handle a much larger hidden-size and negative inputs. Furthermore, results for $\text{NAC}_{\bullet,\sigma}$ and $\text{NAC}_{\bullet,\text{NMU}}$ validate that removing division and adding bias improves the success-rate, but are not enough when the hidden-size is large, as there is no ideal initialization. Interestingly, no models can learn $U[1.1, 1.2]$, suggesting certain input ranges might be troublesome.

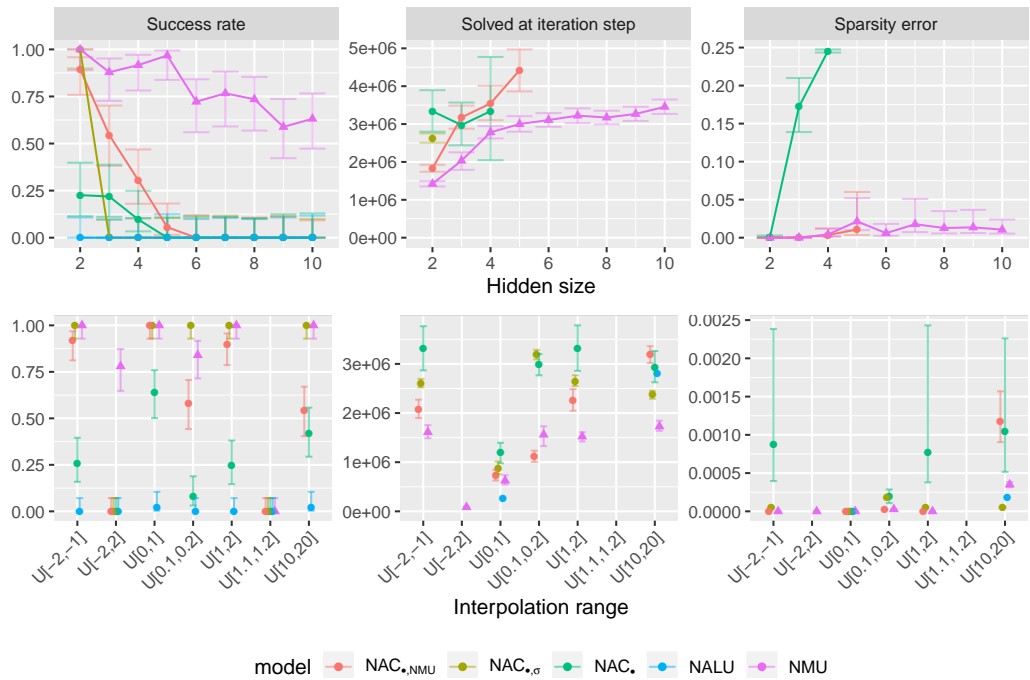

Figure 3: Multiplication task results when varying the hidden input-size and when varying the input-range. Extrapolation ranges are defined in Appendix C.4.

## 4.2 PRODUCT OF SEQUENTIAL MNIST

To investigate if a deep neural network can be optimized when backpropagating through an arithmetic unit, the arithmetic units are used as a recurrent-unit over a sequence of MNIST digits, where the target is to fit the cumulative product. This task is similar to "MNIST Counting and Arithmetic Tasks" in Trask et al. (2018)[2], but uses multiplication rather than addition (addition is in Appendix D.2). Each model is trained on sequences of length 2 and tested on sequences of up to 20 MNIST digits.

We define the success-criterion by comparing the MSE of each model with a baseline model that has a correct solution for the arithmetic unit. If the MSE of each model is less than the upper 1% MSE-confidence-interval of the baseline model, then the model is considered successfully converged.

Sparsity and "solved at iteration step" is determined as described in experiment 4.1. The validation set is the last 5000 MNIST digits from the training set, which is used for early stopping.

In this experiment, we found that having an unconstrained "input-network" can cause the multiplication-units to learn an undesired solution, e.g. $(0.1 \cdot 81 + 1 - 0.1) = 9$. Such network do solve the problem but not in the intended way. To prevent this solution, we regularize the CNN output with $\mathcal{R}_z = \frac{1}{H_{\ell-1} H_\ell} \sum_{h_\ell}^{H_\ell} \sum_{h_{\ell-1}}^{H_{\ell-1}} (1 - W_{h_{\ell-1}, h_\ell}) \cdot (1 - \bar{z}_{h_{\ell-1}})^2$. This regularizer is applied to the NMU and $\mathrm{NAC}_{\bullet,\mathrm{NMU}}$ models. See Appendix D.4 for the results where this regularizer is not used.

Figure 4 shows that the NMU does not hinder learning a more complex neural network. Moreover, the NMU can extrapolate to much longer sequences than what it is trained on.

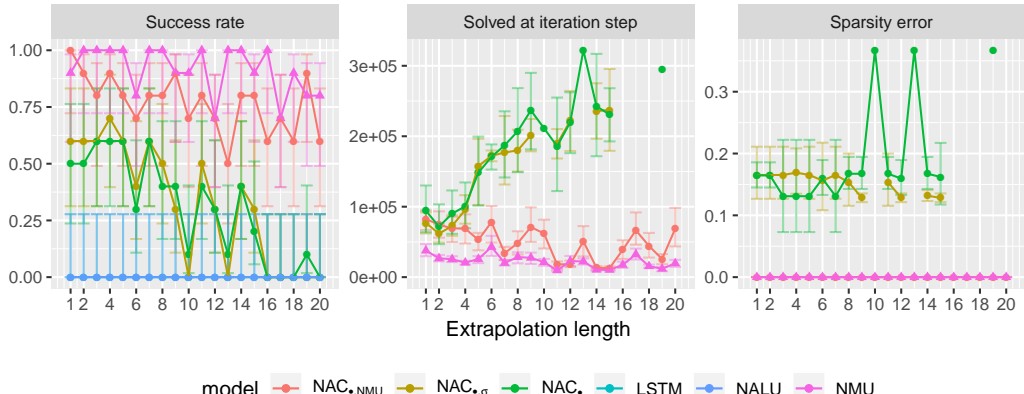

Figure 4: MNIST sequential multiplication task. Each model is trained on sequences of two digits, results are for extrapolating to longer sequences. Error-bars represent the 95% confidence interval.

## 5 CONCLUSION

By including theoretical considerations, such as initialization, gradients, and sparsity, we have developed the Neural Multiplication Unit (NMU) and the Neural Addition Unit (NAU), which outperforms state-of-the-art models on established extrapolation and sequential tasks. Our models converge more consistently, faster, to an more interpretable solution, and supports all input ranges.

A natural next step would be to extend the NMU to support division and add gating between the NMU and NAU, to be comparable in theoretical features with NALU. However we find, both experimentally and theoretically, that learning division is impractical, because of the singularity when dividing by zero, and that a sigmoid-gate choosing between two functions with vastly different convergences properties, such as a multiplication unit and an addition unit, cannot be consistently learned.

Finally, when considering more than just two inputs to the multiplication unit, our model performs significantly better than previously proposed methods and their variations. The ability for a neural unit to consider more than two inputs is critical in neural networks which are often overparameterized.

---

[2]The same CNN is used, `https://github.com/pytorch/examples/tree/master/mnist`.

ACKNOWLEDGMENTS

We would like to thank Andrew Trask and the other authors of the NALU paper, for highlighting the importance and challenges of extrapolation in Neural Networks.

We would also like to thank the students Raja Shan Zaker Kreen and William Frisch Møller from The Technical University of Denmark, who initially showed us that the NALU do not converge consistently.

Alexander R. Johansen and the computing resources from the Technical University of Denmark, where funded by the Innovation Foundation Denmark through the DABAI project.

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

# A  GRADIENT DERIVATIVES

## A.1  WEIGHT MATRIX CONSTRUCTION

For clarity the weight matrix construction is defined using scalar notation

$$W_{h_\ell,h_{\ell-1}} = \tanh(\hat{W}_{h_\ell,h_{\ell-1}})\sigma(\hat{M}_{h_\ell,h_{\ell-1}}) \tag{18}$$

The of the loss with respect to $\hat{W}_{h_\ell,h_{\ell-1}}$ and $\hat{M}_{h_\ell,h_{\ell-1}}$ is then derived using backpropagation.

$$
\begin{aligned}
\frac{\partial \mathcal{L}}{\partial \hat{W}_{h_\ell,h_{\ell-1}}} &= \frac{\partial \mathcal{L}}{\partial W_{h_\ell,h_{\ell-1}}} \frac{\partial W_{h_\ell,h_{\ell-1}}}{\partial \hat{W}_{h_\ell,h_{\ell-1}}} \\
&= \frac{\partial \mathcal{L}}{\partial W_{h_\ell,h_{\ell-1}}} (1 - \tanh^2(\hat{W}_{h_\ell,h_{\ell-1}}))\sigma(\hat{M}_{h_\ell,h_{\ell-1}}) \\
\frac{\partial \mathcal{L}}{\partial \hat{M}_{h_\ell,h_{\ell-1}}} &= \frac{\partial \mathcal{L}}{\partial W_{h_\ell,h_{\ell-1}}} \frac{\partial W_{h_\ell,h_{\ell-1}}}{\partial \hat{M}_{h_\ell,h_{\ell-1}}} \\
&= \frac{\partial \mathcal{L}}{\partial W_{h_\ell,h_{\ell-1}}} \tanh(\hat{W}_{h_\ell,h_{\ell-1}})\sigma(\hat{M}_{h_\ell,h_{\ell-1}})(1 - \sigma(\hat{M}_{h_\ell,h_{\ell-1}}))
\end{aligned}
\tag{19}
$$

As seen from this result, one only needs to consider $\frac{\partial \mathcal{L}}{\partial W_{h_\ell,h_{\ell-1}}}$ for $NAC_+$ and $NAC_\bullet$, as the gradient with respect to $\hat{W}_{h_\ell,h_{\ell-1}}$ and $\hat{M}_{h_\ell,h_{\ell-1}}$ is a multiplication on $\frac{\partial \mathcal{L}}{\partial W_{h_\ell,h_{\ell-1}}}$.

## A.2  GRADIENT OF $NAC_\bullet$

The $NAC_\bullet$ is defined using scalar notation.

$$z_{h_\ell} = \exp\left(\sum_{h_{\ell-1}=1}^{H_{\ell-1}} W_{h_\ell,h_{\ell-1}} \log(|z_{h_{\ell-1}}| + \epsilon)\right) \tag{20}$$

The gradient of the loss with respect to $W_{h_\ell,h_{\ell-1}}$ can the be derived using backpropagation.

$$
\begin{aligned}
\frac{\partial z_{h_\ell}}{\partial W_{h_\ell,h_{\ell-1}}} &= \exp\left(\sum_{h'_{\ell-1}=1}^{H_{\ell-1}} W_{h_\ell,h'_{\ell-1}} \log(|z_{h'_{\ell-1}}| + \epsilon)\right) \log(|z_{h_{\ell-1}}| + \epsilon) \\
&= z_{h_\ell} \log(|z_{h_{\ell-1}}| + \epsilon)
\end{aligned}
\tag{21}
$$

We now wish to derive the backpropagation term $\delta_{h_\ell} = \frac{\partial \mathcal{L}}{\partial z_{h_\ell}}$, because $z_{h_\ell}$ affects $\{z_{h_{\ell+1}}\}_{h_{\ell+1}=1}^{H_{\ell+1}}$ this becomes:

$$\delta_{h_\ell} = \frac{\partial \mathcal{L}}{\partial z_{h_\ell}} = \sum_{h_{\ell+1}=1}^{H_{\ell+1}} \frac{\partial \mathcal{L}}{\partial z_{h_{\ell+1}}} \frac{\partial z_{h_{\ell+1}}}{\partial z_{h_\ell}} = \sum_{h_{\ell+1}=1}^{H_{\ell+1}} \delta_{h_{\ell+1}} \frac{\partial z_{h_{\ell+1}}}{\partial z_{h_\ell}} \tag{22}$$

To make it easier to derive $\frac{\partial z_{h_{\ell+1}}}{\partial z_{h_\ell}}$ we re-express the $z_{h_\ell}$ as $z_{h_{\ell+1}}$.

$$z_{h_{\ell+1}} = \exp\left(\sum_{h_\ell=1}^{H_\ell} W_{h_{\ell+1},h_\ell} \log(|z_{h_\ell}| + \epsilon)\right) \tag{23}$$

The gradient of $\frac{\partial z_{h_{\ell+1}}}{\partial z_{h_\ell}}$ is then:

$$
\begin{aligned}
\frac{\partial z_{h_{\ell+1}}}{\partial z_{h_\ell}} &= \exp\left(\sum_{h_\ell=1}^{H_\ell} W_{h_{\ell+1},h_\ell} \log(|z_{h_\ell}| + \epsilon)\right) W_{h_{\ell+1},h_\ell} \frac{\partial \log(|z_{h_\ell}| + \epsilon)}{\partial z_{h_\ell}} \\
&= \exp\left(\sum_{h_\ell=1}^{H_\ell} W_{h_{\ell+1},h_\ell} \log(|z_{h_\ell}| + \epsilon)\right) W_{h_{\ell+1},h_\ell} \frac{\mathrm{abs}'(z_{h_\ell})}{|z_{h_\ell}| + \epsilon} \\
&= z_{h_{\ell+1}} W_{h_{\ell+1},h_\ell} \frac{\mathrm{abs}'(z_{h_\ell})}{|z_{h_\ell}| + \epsilon}
\end{aligned}
\tag{24}
$$

$\mathrm{abs}'(z_{h_\ell})$ is the gradient of the absolute function. In the paper we denote this as $\mathrm{sign}(z_{h_\ell})$ for brevity. However, depending on the exact definition used there may be a difference for $z_{h_\ell} = 0$, as $\mathrm{abs}'(0)$ is undefined. In practicality this doesn't matter much though, although theoretically it does mean that the expectation of this is theoretically undefined when $E[z_{h_\ell}] = 0$.

### A.3 Gradient of NMU

In scalar notation the NMU is defined as:

$$
z_{h_\ell} = \prod_{h_{\ell-1}=1}^{H_{\ell-1}} \left(W_{h_{\ell-1},h_\ell} z_{h_{\ell-1}} + 1 - W_{h_{\ell-1},h_\ell}\right)
\tag{25}
$$

The gradient of the loss with respect to $W_{h_{\ell-1},h_\ell}$ is fairly trivial. Note that every term but the one for $h_{\ell-1}$, is just a constant with respect to $W_{h_{\ell-1},h_\ell}$. The product, except the term for $h_{\ell-1}$ can be expressed as $\frac{z_{h_\ell}}{W_{h_{\ell-1},h_\ell} z_{h_{\ell-1}} + 1 - W_{h_{\ell-1},h_\ell}}$. Using this fact, the gradient can be expressed as:

$$
\frac{\partial \mathcal{L}}{\partial w_{h_\ell,h_{\ell-1}}} = \frac{\partial \mathcal{L}}{\partial z_{h_\ell}} \frac{\partial z_{h_\ell}}{\partial w_{h_\ell,h_{\ell-1}}} = \frac{\partial \mathcal{L}}{\partial z_{h_\ell}} \frac{z_{h_\ell}}{W_{h_{\ell-1},h_\ell} z_{h_{\ell-1}} + 1 - W_{h_{\ell-1},h_\ell}} \left(z_{h_{\ell-1}} - 1\right)
\tag{26}
$$

Similarly, the gradient $\frac{\partial \mathcal{L}}{\partial z_{h_\ell}}$ which is essential in backpropagation can equally easily be derived as:

$$
\frac{\partial \mathcal{L}}{\partial z_{h_{\ell-1}}} = \sum_{h_\ell=1}^{H_\ell} \frac{\partial \mathcal{L}}{\partial z_{h_\ell}} \frac{\partial z_{h_\ell}}{\partial z_{h_{\ell-1}}} = \sum_{h_\ell=1}^{H_\ell} \frac{z_{h_\ell}}{W_{h_{\ell-1},h_\ell} z_{h_{\ell-1}} + 1 - W_{h_{\ell-1},h_\ell}} W_{h_{\ell-1},h_\ell}
\tag{27}
$$

# B  MOMENTS

## B.1  OVERVIEW

### B.1.1  MOMENTS AND INITIALIZATION FOR ADDITION

The desired properties for initialization are according to Glorot et al. (Glorot & Bengio, 2010):

$$E[z_{h_\ell}] = 0 \qquad\qquad E\left[\frac{\partial\mathcal{L}}{\partial z_{h_{\ell-1}}}\right] = 0$$
$$Var[z_{h_\ell}] = Var\left[z_{h_{\ell-1}}\right] \quad Var\left[\frac{\partial\mathcal{L}}{\partial z_{h_{\ell-1}}}\right] = Var\left[\frac{\partial\mathcal{L}}{\partial z_{h_\ell}}\right] \tag{28}$$

### B.1.2  INITIALIZATION FOR ADDITION

Glorot initialization can not be used for $\text{NAC}_+$ as $W_{h_{\ell-1},h_\ell}$ is not sampled directly. Assuming that $\hat{W}_{h_\ell,h_{\ell-1}} \sim \text{Uniform}[-r,r]$ and $\hat{M}_{h_\ell,h_{\ell-1}} \sim \text{Uniform}[-r,r]$, then the variance can be derived (see proof in Appendix B.2) to be:

$$Var[W_{h_{\ell-1},h_\ell}] = \frac{1}{2r}\left(1 - \frac{\tanh(r)}{r}\right)\left(r - \tanh\left(\frac{r}{2}\right)\right) \tag{29}$$

One can then solve for $r$, given the desired variance ($Var[W_{h_{\ell-1},h_\ell}] = \frac{2}{H_{\ell-1}+H_\ell}$) (Glorot & Bengio, 2010).

### B.1.3  MOMENTS AND INITIALIZATION FOR MULTIPLICATION

Using second order multivariate Taylor approximation and some assumptions of uncorrelated stochastic variables, the expectation and variance of the $\text{NAC}_\bullet$ layer can be estimated to:

$$f(c_1, c_2) = \left(1 + c_1 \frac{1}{2} Var[W_{h_\ell,h_{\ell-1}}] \log(|E[z_{h_{\ell-1}}]| + \epsilon)^2\right)^{c_2 \; H_{\ell-1}}$$
$$E[z_{h_\ell}] \approx f(1,1)$$
$$Var[z_{h_\ell}] \approx f(4,1) - f(1,2)$$
$$E\left[\frac{\partial\mathcal{L}}{\partial z_{h_{\ell-1}}}\right] = 0 \tag{30}$$
$$Var\left[\frac{\partial\mathcal{L}}{\partial z_{h_{\ell-1}}}\right] \approx Var\left[\frac{\partial\mathcal{L}}{\partial z_{h_\ell}}\right] H_\ell \; f(4,1) \; Var[W_{h_\ell,h_{\ell-1}}]$$
$$\cdot \left(\frac{1}{\left(|E[z_{h_{\ell-1}}]| + \epsilon\right)^2} + \frac{3}{\left(|E[z_{h_{\ell-1}}]| + \epsilon\right)^4} Var[z_{h_{\ell-1}}]\right)$$

This is problematic because $E[z_{h_\ell}] \geq 1$, and the variance explodes for $E[z_{h_{\ell-1}}] = 0$. $E[z_{h_{\ell-1}}] = 0$ is normally a desired property (Glorot & Bengio, 2010). The variance explodes for $E[z_{h_{\ell-1}}] = 0$, and can thus not be initialized to anything meaningful.

For our proposed NMU, the expectation and variance can be derived (see proof in Appendix B.4) using the same assumptions as before, although no Taylor approximation is required:

$$E[z_{h_\ell}] \approx \left(\frac{1}{2}\right)^{H_{\ell-1}}$$

$$E\left[\frac{\partial \mathcal{L}}{\partial z_{h_{\ell-1}}}\right] \approx 0$$

$$Var[z_{h_\ell}] \approx \left(Var[W_{h_{\ell-1},h_\ell}] + \frac{1}{4}\right)^{H_{\ell-1}} \left(Var[z_{h_{\ell-1}}] + 1\right)^{H_{\ell-1}} - \left(\frac{1}{4}\right)^{H_{\ell-1}} \quad (31)$$

$$Var\left[\frac{\partial \mathcal{L}}{\partial z_{h_{\ell-1}}}\right] \approx Var\left[\frac{\partial \mathcal{L}}{\partial z_{h_\ell}}\right] H_\ell$$
$$\cdot \left(\left(Var[W_{h_{\ell-1},h_\ell}] + \frac{1}{4}\right)^{H_{\ell-1}} \left(Var[z_{h_{\ell-1}}] + 1\right)^{H_{\ell-1}-1}\right)$$

These expectations are better behaved. It is unlikely that the expectation of a multiplication unit can become zero, since the identity for multiplication is 1. However, for a large $H_{\ell-1}$ it will be near zero.

The variance is also better behaved, but do not provide a input-independent initialization strategy. We propose initializing with $Var[W_{h_{\ell-1},h_\ell}] = \frac{1}{4}$, as this is the solution to $Var[z_{h_\ell}] = Var[z_{h_{\ell-1}}]$ assuming $Var[z_{h_{\ell-1}}] = 1$ and a large $H_{\ell-1}$ (see proof in Appendix B.4.3). However, more exact solutions are possible if the input variance is known.

## B.2 Expectation and variance for weight matrix construction in NAC layers

The weight matrix construction in NAC, is defined in scalar notation as:

$$W_{h_\ell,h_{\ell-1}} = \tanh(\hat{W}_{h_\ell,h_{\ell-1}})\sigma(\hat{M}_{h_\ell,h_{\ell-1}}) \quad (32)$$

Simplifying the notation of this, and re-expressing it using stochastic variables with uniform distributions this can be written as:

$$W \sim \tanh(\hat{W})\sigma(\hat{M})$$
$$\hat{W} \sim U[-r,r] \quad (33)$$
$$\hat{M} \sim U[-r,r]$$

Since $\tanh(\hat{W})$ is an odd-function and $E[\hat{W}] = 0$, deriving the expectation $E[W]$ is trivial.

$$E[W] = E[\tanh(\hat{W})]E[\sigma(\hat{M})] = 0 \cdot E[\sigma(\hat{M})] = 0 \quad (34)$$

The variance is more complicated, however as $\hat{W}$ and $\hat{M}$ are independent, it can be simplified to:

$$Var[W] = E[\tanh(\hat{W})^2]E[\sigma(\hat{M})^2] - E[\tanh(\hat{W})]^2E[\sigma(\hat{M})]^2 = E[\tanh(\hat{W})^2]E[\sigma(\hat{M})^2] \quad (35)$$

These second moments can be analyzed independently. First for $E[\tanh(\hat{W})^2]$:

$$E[\tanh(\hat{W})^2] = \int_{-\infty}^{\infty} \tanh(x)^2 f_{U[-r,r]}(x) \, dx$$
$$= \frac{1}{2r} \int_{-r}^{r} \tanh(x)^2 \, dx$$
$$= \frac{1}{2r} \cdot 2 \cdot (r - \tanh(r)) \quad (36)$$
$$= 1 - \frac{\tanh(r)}{r}$$

Then for $E[\tanh(\hat{M})^2]$:

$$
\begin{aligned}
E[\sigma(\hat{M})^2] &= \int_{-\infty}^{\infty} \sigma(x)^2 f_{U[-r,r]}(x)\, dx \\
&= \frac{1}{2r} \int_{-r}^{r} \sigma(x)^2\, dx \\
&= \frac{1}{2r} \left( r - \tanh\left(\frac{r}{2}\right) \right)
\end{aligned}
\tag{37}
$$

Which results in the variance:

$$
\mathrm{Var}[W] = \frac{1}{2r} \left( 1 - \frac{\tanh(r)}{r} \right) \left( r - \tanh\left(\frac{r}{2}\right) \right)
\tag{38}
$$

## B.3 EXPECTATION AND VARIANCE OF NAC$_\bullet$

### B.3.1 FORWARD PASS

**Expectation**  Assuming that each $z_{h_{\ell-1}}$ are uncorrelated, the expectation can be simplified to:

$$
\begin{aligned}
E[z_{h_\ell}] &= E\left[ \exp\left( \sum_{h_{\ell-1}=1}^{H_{\ell-1}} W_{h_\ell,h_{\ell-1}} \log(|z_{h_{\ell-1}}| + \epsilon) \right) \right] \\
&= E\left[ \prod_{h_{\ell-1}=1}^{H_{\ell-1}} \exp(W_{h_\ell,h_{\ell-1}} \log(|z_{h_{\ell-1}}| + \epsilon)) \right] \\
&\approx \prod_{h_{\ell-1}=1}^{H_{\ell-1}} E[\exp(W_{h_\ell,h_{\ell-1}} \log(|z_{h_{\ell-1}}| + \epsilon))] \\
&= E[\exp(W_{h_\ell,h_{\ell-1}} \log(|z_{h_{\ell-1}}| + \epsilon))]^{H_{\ell-1}} \\
&= E\left[ (|z_{h_{\ell-1}}| + \epsilon)^{W_{h_\ell,h_{\ell-1}}} \right]^{H_{\ell-1}} \\
&= E\left[ f(z_{h_{\ell-1}}, W_{h_\ell,h_{\ell-1}}) \right]^{H_{\ell-1}}
\end{aligned}
\tag{39}
$$

Here we define $g$ as a non-linear transformation function of two independent stochastic variables:

$$
f(z_{h_{\ell-1}}, W_{h_\ell,h_{\ell-1}}) = (|z_{h_{\ell-1}}| + \epsilon)^{W_{h_\ell,h_{\ell-1}}}
\tag{40}
$$

We then apply second order Taylor approximation of $f$, around $(E[z_{h_{\ell-1}}], E[W_{h_\ell,h_{\ell-1}}])$.

$$
\begin{aligned}
E[f(z_{h_{\ell-1}}, W_{h_\ell,h_{\ell-1}})] \approx E\Bigg[ & \\
& f(E[z_{h_{\ell-1}}], E[W_{h_\ell,h_{\ell-1}}]) \\
& + \begin{bmatrix} z_{h_{\ell-1}} - E[z_{h_{\ell-1}}] \\ W_{h_\ell,h_{\ell-1}} - E[W_{h_\ell,h_{\ell-1}}] \end{bmatrix}^T \begin{bmatrix} \frac{\partial f(z_{h_{\ell-1}}, W_{h_\ell,h_{\ell-1}})}{\partial z_{h_{\ell-1}}} \\ \frac{\partial f(z_{h_{\ell-1}}, W_{h_\ell,h_{\ell-1}})}{\partial W_{h_\ell,h_{\ell-1}}} \end{bmatrix} \Bigg|_{\begin{cases} z_{h_{\ell-1}} = E[z_{h_{\ell-1}}] \\ W_{h_\ell,h_{\ell-1}} = E[W_{h_\ell,h_{\ell-1}}] \end{cases}} \\
& + \frac{1}{2} \begin{bmatrix} z_{h_{\ell-1}} - E[z_{h_{\ell-1}}] \\ W_{h_\ell,h_{\ell-1}} - E[W_{h_\ell,h_{\ell-1}}] \end{bmatrix}^T \\
& \bullet \begin{bmatrix} \frac{\partial^2 f(z_{h_{\ell-1}}, W_{h_\ell,h_{\ell-1}})}{\partial^2 z_{h_{\ell-1}}} & \frac{\partial^2 f(z_{h_{\ell-1}}, W_{h_\ell,h_{\ell-1}})}{\partial z_{h_{\ell-1}} \partial W_{h_\ell,h_{\ell-1}}} \\ \frac{\partial^2 f(z_{h_{\ell-1}}, W_{h_\ell,h_{\ell-1}})}{\partial z_{h_{\ell-1}} \partial W_{h_\ell,h_{\ell-1}}} & \frac{\partial^2 f(z_{h_{\ell-1}}, W_{h_\ell,h_{\ell-1}})}{\partial^2 W_{h_\ell,h_{\ell-1}}} \end{bmatrix} \Bigg|_{\begin{cases} z_{h_{\ell-1}} = E[z_{h_{\ell-1}}] \\ W_{h_\ell,h_{\ell-1}} = E[W_{h_\ell,h_{\ell-1}}] \end{cases}} \\
& \bullet \begin{bmatrix} z_{h_{\ell-1}} - E[z_{h_{\ell-1}}] \\ W_{h_\ell,h_{\ell-1}} - E[W_{h_\ell,h_{\ell-1}}] \end{bmatrix} \Bigg]
\end{aligned}
\tag{41}
$$

Because $E[z_{h_{\ell-1}} - E[z_{h_{\ell-1}}]] = 0$, $E[W_{h_\ell,h_{\ell-1}} - E[W_{h_\ell,h_{\ell-1}}]] = 0$, and $Cov[z_{h_{\ell-1}}, W_{h_\ell,h_{\ell-1}}] = 0$. This simplifies to:

$$
\begin{aligned}
E[g(z_{h_{\ell-1}}, W_{h_\ell,h_{\ell-1}})] &\approx g(E[z_{h_{\ell-1}}], E[W_{h_\ell,h_{\ell-1}}]) \\
&+ \frac{1}{2} Var \begin{bmatrix} z_{h_{\ell-1}} \\ W_{h_\ell,h_{\ell-1}} \end{bmatrix}^T \begin{bmatrix} \frac{\partial^2 g(z_{h_{\ell-1}}, W_{h_\ell,h_{\ell-1}})}{\partial^2 z_{h_{\ell-1}}} \\ \frac{\partial^2 g(z_{h_{\ell-1}}, W_{h_\ell,h_{\ell-1}})}{\partial^2 W_{h_\ell,h_{\ell-1}}} \end{bmatrix} \Bigg| \begin{cases} z_{h_{\ell-1}} = E[z_{h_{\ell-1}}] \\ W_{h_\ell,h_{\ell-1}} = E[W_{h_\ell,h_{\ell-1}}] \end{cases}
\end{aligned}
\tag{42}
$$

Inserting the derivatives and computing the inner products yields:

$$
\begin{aligned}
E[f(z_{h_{\ell-1}}, W_{h_\ell,h_{\ell-1}})] &\approx (|E[z_{h_{\ell-1}}]| + \epsilon)^{E[W_{h_\ell,h_{\ell-1}}]} \\
&+ \frac{1}{2} Var[z_{h_{\ell-1}}](|E[z_{h_{\ell-1}}]| + \epsilon)^{E[W_{h_\ell,h_{\ell-1}}]-2} E[W_{h_\ell,h_{\ell-1}}](E[W_{h_\ell,h_{\ell-1}}] - 1) \\
&+ \frac{1}{2} Var[W_{h_\ell,h_{\ell-1}}](|E[z_{h_{\ell-1}}]| + \epsilon)^{E[W_{h_\ell,h_{\ell-1}}]} \log(|E[z_{h_{\ell-1}}]| + \epsilon)^2 \\
&= 1 + \frac{1}{2} Var[W_{h_\ell,h_{\ell-1}}] \log(|E[z_{h_{\ell-1}}]| + \epsilon)^2
\end{aligned}
\tag{43}
$$

This gives the final expectation:

$$
\begin{aligned}
E[z_{h_\ell}] &= E\left[g(z_{h_{\ell-1}}, W_{h_\ell,h_{\ell-1}})\right]^{H_{\ell-1}} \\
&\approx \left(1 + \frac{1}{2} Var[W_{h_\ell,h_{\ell-1}}] \log(|E[z_{h_{\ell-1}}]| + \epsilon)^2\right)^{H_{\ell-1}}
\end{aligned}
\tag{44}
$$

We evaluate the error of the approximation, where $W_{h_\ell,h_{\ell-1}} \sim U[-r_w, r_w]$ and $z_{h_{\ell-1}} \sim U[0, r_z]$. These distributions are what is used in the arithmetic dataset. The error is plotted in figure 5.

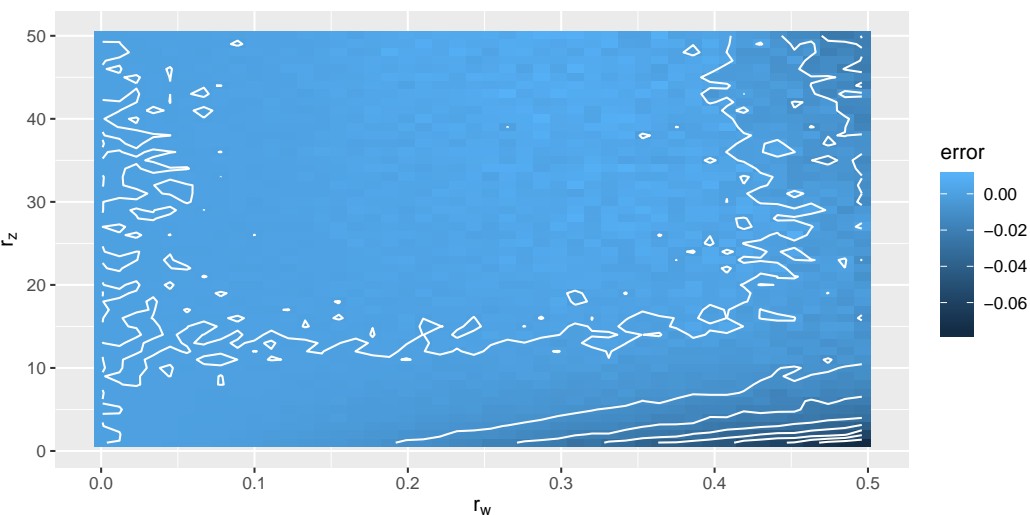

Figure 5: Error between theoretical approximation and the numerical approximation estimated by random sampling of 100000 observations at each combination of $r_z$ and $r_w$.

**Variance** The variance can be derived using the same assumptions as used in "expectation", that all $z_{h_{\ell-1}}$ are uncorrelated.

$$
\begin{aligned}
Var[z_{h_\ell}] &= E[z_{h_\ell}^2] - E[z_{h_\ell}]^2 \\
&= E\left[\prod_{h_{\ell-1}=1}^{H_{\ell-1}} (|z_{h_{\ell-1}}| + \epsilon)^{2 \cdot W_{h_\ell,h_{\ell-1}}}\right] - E\left[\prod_{h_{\ell-1}=1}^{H_{\ell-1}} (|z_{h_{\ell-1}}| + \epsilon)^{W_{h_\ell,h_{\ell-1}}}\right]^2 \\
&= E\left[f(z_{h_{\ell-1}}, 2 \cdot W_{h_\ell,h_{\ell-1}})\right]^{H_{\ell-1}} - E\left[f(z_{h_{\ell-1}}, W_{h_\ell,h_{\ell-1}})\right]^{2 \cdot H_{\ell-1}}
\end{aligned}
\tag{45}
$$

We already have from the expectation result in (43) that:

$$E\left[f(z_{h_{\ell-1}}, W_{h_\ell,h_{\ell-1}})\right] \approx 1 + \frac{1}{2}Var[W_{h_\ell,h_{\ell-1}}]\log(|E[z_{h_{\ell-1}}]| + \epsilon)^2 \tag{46}$$

By substitution of variable we have that:

$$E\left[f(z_{h_{\ell-1}}, 2\cdot W_{h_\ell,h_{\ell-1}})\right] \approx 1 + \frac{1}{2}Var[2\cdot W_{h_\ell,h_{\ell-1}}]\log(|E[z_{h_{\ell-1}}]| + \epsilon)^2$$
$$\approx 1 + 2\cdot Var[W_{h_\ell,h_{\ell-1}}]\log(|E[z_{h_{\ell-1}}]| + \epsilon)^2 \tag{47}$$

This gives the variance:

$$Var[z_{h_\ell}] = E\left[g(z_{h_{\ell-1}}, 2\cdot W_{h_\ell,h_{\ell-1}})\right]^{H_{\ell-1}} - E\left[f(z_{h_{\ell-1}}, W_{h_\ell,h_{\ell-1}})\right]^{2\cdot H_{\ell-1}}$$
$$\approx \left(1 + 2\cdot Var[W_{h_\ell,h_{\ell-1}}]\log(|E[z_{h_{\ell-1}}]| + \epsilon)^2\right)^{H_{\ell-1}} \tag{48}$$
$$- \left(1 + \frac{1}{2}\cdot Var[W_{h_\ell,h_{\ell-1}}]\log(|E[z_{h_{\ell-1}}]| + \epsilon)^2\right)^{2\cdot H_{\ell-1}}$$

### B.3.2 BACKWARD PASS

**Expectation** The expectation of the back-propagation term assuming that $\delta_{h_{\ell+1}}$ and $\frac{\partial z_{h_{\ell+1}}}{\partial z_{h_\ell}}$ are mutually uncorrelated:

$$E[\delta_{h_\ell}] = E\left[\sum_{h_{\ell+1}=1}^{H_{\ell+1}} \delta_{h_{\ell+1}}\frac{\partial z_{h_{\ell+1}}}{\partial z_{h_\ell}}\right] \approx H_{\ell+1}E[\delta_{h_{\ell+1}}]E\left[\frac{\partial z_{h_{\ell+1}}}{\partial z_{h_\ell}}\right] \tag{49}$$

Assuming that $z_{h_{\ell+1}}$, $W_{h_{\ell+1},h_\ell}$, and $z_{h_\ell}$ are uncorrelated:

$$E\left[\frac{\partial z_{h_{\ell+1}}}{\partial z_{h_\ell}}\right] \approx E[z_{h_{\ell+1}}]E[W_{h_{\ell+1},h_\ell}]E\left[\frac{\mathrm{abs}'(z_{h_\ell})}{|z_{h_\ell}| + \epsilon}\right] = E[z_{h_{\ell+1}}]\cdot 0\cdot E\left[\frac{\mathrm{abs}'(z_{h_\ell})}{|z_{h_\ell}| + \epsilon}\right] = 0 \tag{50}$$

**Variance** Deriving the variance is more complicated:

$$Var\left[\frac{\partial z_{h_{\ell+1}}}{\partial z_{h_\ell}}\right] = Var\left[z_{h_{\ell+1}}W_{h_{\ell+1},h_\ell}\frac{\mathrm{abs}'(z_{h_\ell})}{|z_{h_\ell}| + \epsilon}\right] \tag{51}$$

Assuming again that $z_{h_{\ell+1}}$, $W_{h_{\ell+1},h_\ell}$, and $z_{h_\ell}$ are uncorrelated, and likewise for their second moment:

$$Var\left[\frac{\partial z_{h_{\ell+1}}}{\partial z_{h_\ell}}\right] \approx E[z_{h_{\ell+1}}^2]E[W_{h_{\ell+1},h_\ell}^2]E\left[\left(\frac{\mathrm{abs}'(z_{h_\ell})}{|z_{h_\ell}| + \epsilon}\right)^2\right]$$
$$- E[z_{h_{\ell+1}}]^2E[W_{h_{\ell+1},h_\ell}]^2E\left[\frac{\mathrm{abs}'(z_{h_\ell})}{|z_{h_\ell}| + \epsilon}\right]^2$$
$$= E[z_{h_{\ell+1}}^2]Var[W_{h_{\ell+1},h_\ell}]E\left[\left(\frac{\mathrm{abs}'(z_{h_\ell})}{|z_{h_\ell}| + \epsilon}\right)^2\right] \tag{52}$$
$$- E[z_{h_{\ell+1}}]^2\cdot 0\cdot E\left[\frac{\mathrm{abs}'(z_{h_\ell})}{|z_{h_\ell}| + \epsilon}\right]^2$$
$$= E[z_{h_{\ell+1}}^2]Var[W_{h_{\ell+1},h_\ell}]E\left[\left(\frac{\mathrm{abs}'(z_{h_\ell})}{|z_{h_\ell}| + \epsilon}\right)^2\right]$$

Using Taylor approximation around $E[z_{h_\ell}]$ we have:

$$E\left[\left(\frac{\mathrm{abs}'(z_{h_\ell})}{|z| + \epsilon}\right)^2\right] \approx \frac{1}{(|E[z_{h_\ell}]| + \epsilon)^2} + \frac{1}{2}\frac{6}{(|E[z_{h_\ell}]| + \epsilon)^4}Var[z_{h_\ell}]$$
$$= \frac{1}{(|E[z_{h_\ell}]| + \epsilon)^2} + \frac{3}{(|E[z_{h_\ell}]| + \epsilon)^4}Var[z_{h_\ell}] \tag{53}$$

Finally, by reusing the result for $E[z_{h_\ell}^2]$ from earlier the variance can be expressed as:

$$
Var\left[\frac{\partial \mathcal{L}}{\partial z_{h_{\ell-1}}}\right] \approx Var\left[\frac{\partial \mathcal{L}}{\partial z_{h_\ell}}\right] H_\ell \left(1 + 2 \cdot Var[W_{h_\ell,h_{\ell-1}}] \log(|E[z_{h_{\ell-1}}]| + \epsilon)^2\right)^{H_{\ell-1}}
$$
$$
\cdot Var[W_{h_\ell,h_{\ell-1}}] \left(\frac{1}{\left(|E[z_{h_{\ell-1}}]| + \epsilon\right)^2} + \frac{3}{\left(|E[z_{h_{\ell-1}}]| + \epsilon\right)^4} Var[z_{h_{\ell-1}}]\right) \tag{54}
$$

### B.4 EXPECTATION AND VARIANCE OF NMU

#### B.4.1 FORWARD PASS

**Expectation** Assuming that all $z_{h_{\ell-1}}$ are independent:

$$
E[z_{h_\ell}] = E\left[\prod_{h_{\ell-1}=1}^{H_{\ell-1}} \left(W_{h_{\ell-1},h_\ell} z_{h_{\ell-1}} + 1 - W_{h_{\ell-1},h_\ell}\right)\right]
$$
$$
\approx E\left[W_{h_{\ell-1},h_\ell} z_{h_{\ell-1}} + 1 - W_{h_{\ell-1},h_\ell}\right]^{H_{\ell-1}} \tag{55}
$$
$$
\approx \left(E[W_{h_{\ell-1},h_\ell}]E[z_{h_{\ell-1}}] + 1 - E[W_{h_{\ell-1},h_\ell}]\right)^{H_{\ell-1}}
$$

Assuming that $E[z_{h_{\ell-1}}] = 0$ which is a desired property and initializing $E[W_{h_{\ell-1},h_\ell}] = 1/2$, the expectation is:

$$
E[z_{h_\ell}] \approx \left(E[W_{h_{\ell-1},h_\ell}]E[z_{h_{\ell-1}}] + 1 - E[W_{h_{\ell-1},h_\ell}]\right)^{H_{\ell-1}}
$$
$$
\approx \left(\frac{1}{2} \cdot 0 + 1 - \frac{1}{2}\right)^{H_{\ell-1}} \tag{56}
$$
$$
= \left(\frac{1}{2}\right)^{H_{\ell-1}}
$$

**Variance** Reusing the result for the expectation, assuming again that all $z_{h_{\ell-1}}$ are uncorrelated, and using the fact that $W_{h_{\ell-1},h_\ell}$ is initially independent from $z_{h_{\ell-1}}$:

$$
Var[z_{h_\ell}] = E[z_{h_\ell}^2] - E[z_{h_\ell}]^2
$$
$$
\approx E[z_{h_\ell}^2] - \left(\frac{1}{2}\right)^{2 \cdot H_{\ell-1}}
$$
$$
= E\left[\prod_{h_{\ell-1}=1}^{H_{\ell-1}} \left(W_{h_{\ell-1},h_\ell} z_{h_{\ell-1}} + 1 - W_{h_{\ell-1},h_\ell}\right)^2\right] - \left(\frac{1}{2}\right)^{2 \cdot H_{\ell-1}}
$$
$$
\approx E[\left(W_{h_{\ell-1},h_\ell} z_{h_{\ell-1}} + 1 - W_{h_{\ell-1},h_\ell}\right)^2]^{H_{\ell-1}} - \left(\frac{1}{2}\right)^{2 \cdot H_{\ell-1}} \tag{57}
$$
$$
= \left(E[W_{h_{\ell-1},h_\ell}^2]E[z_{h_{\ell-1}}^2] - 2E[W_{h_{\ell-1},h_\ell}^2]E[z_{h_{\ell-1}}] + E[W_{h_{\ell-1},h_\ell}^2]\right.
$$
$$
\left. + 2E[W_{h_{\ell-1},h_\ell}]E[z_{h_{\ell-1}}] - 2E[W_{h_{\ell-1},h_\ell}] + 1\right)^{H_{\ell-1}} - \left(\frac{1}{2}\right)^{2 \cdot H_{\ell-1}}
$$

Assuming that $E[z_{h_{\ell-1}}] = 0$, which is a desired property and initializing $E[W_{h_{\ell-1},h_\ell}] = 1/2$, the variance becomes:

$$
Var[z_{h_\ell}] \approx \left(E[W_{h_{\ell-1},h_\ell}^2]\left(E[z_{h_{\ell-1}}^2] + 1\right)\right)^{H_{\ell-1}} - \left(\frac{1}{2}\right)^{2 \cdot H_{\ell-1}}
$$
$$
\approx \left(\left(Var[W_{h_{\ell-1},h_\ell}] + E[W_{h_{\ell-1},h_\ell}]^2\right)\left(Var[z_{h_{\ell-1}}] + 1\right)\right)^{H_{\ell-1}} - \left(\frac{1}{2}\right)^{2 \cdot H_{\ell-1}} \tag{58}
$$
$$
= \left(Var[W_{h_{\ell-1},h_\ell}] + \frac{1}{4}\right)^{H_{\ell-1}} \left(Var[z_{h_{\ell-1}}] + 1\right)^{H_{\ell-1}} - \left(\frac{1}{2}\right)^{2 \cdot H_{\ell-1}}
$$

### B.4.2 BACKWARD PASS

**Expectation** For the backward pass the expectation can, assuming that $\frac{\partial \mathcal{L}}{\partial z_{h_\ell}}$ and $\frac{\partial z_{h_\ell}}{\partial z_{h_{\ell-1}}}$ are uncorrelated, be derived to:

$$
\begin{aligned}
E\left[\frac{\partial \mathcal{L}}{\partial z_{h_{\ell-1}}}\right] &= H_\ell E\left[\frac{\partial \mathcal{L}}{\partial z_{h_\ell}} \frac{\partial z_{h_\ell}}{\partial z_{h_{\ell-1}}}\right] \\
&\approx H_\ell E\left[\frac{\partial \mathcal{L}}{\partial z_{h_\ell}}\right] E\left[\frac{\partial z_{h_\ell}}{\partial z_{h_{\ell-1}}}\right] \\
&= H_\ell E\left[\frac{\partial \mathcal{L}}{\partial z_{h_\ell}}\right] E\left[\frac{z_{h_\ell}}{W_{h_{\ell-1},h_\ell} z_{h_{\ell-1}} + 1 - W_{h_{\ell-1},h_\ell}} W_{h_{\ell-1},h_\ell}\right] \\
&= H_\ell E\left[\frac{\partial \mathcal{L}}{\partial z_{h_\ell}}\right] E\left[\frac{z_{h_\ell}}{W_{h_{\ell-1},h_\ell} z_{h_{\ell-1}} + 1 - W_{h_{\ell-1},h_\ell}}\right] E\left[W_{h_{\ell-1},h_\ell}\right]
\end{aligned}
\tag{59}
$$

Initializing $E[W_{h_{\ell-1},h_\ell}] = 1/2$, and inserting the result for the expectation $E\left[\frac{z_{h_\ell}}{W_{h_{\ell-1},h_\ell} z_{h_{\ell-1}} + 1 - W_{h_{\ell-1},h_\ell}}\right]$.

$$
\begin{aligned}
E\left[\frac{\partial \mathcal{L}}{\partial z_{h_{\ell-1}}}\right] &\approx H_\ell E\left[\frac{\partial \mathcal{L}}{\partial z_{h_\ell}}\right] \left(\frac{1}{2}\right)^{H_{\ell-1}-1} \frac{1}{2} \\
&= E\left[\frac{\partial \mathcal{L}}{\partial z_{h_\ell}}\right] H_\ell \left(\frac{1}{2}\right)^{H_{\ell-1}}
\end{aligned}
\tag{60}
$$

Assuming that $E\left[\frac{\partial \mathcal{L}}{\partial z_{h_\ell}}\right] = 0$, which is a desired property (Glorot & Bengio, 2010).

$$
\begin{aligned}
E\left[\frac{\partial \mathcal{L}}{\partial z_{h_{\ell-1}}}\right] &\approx 0 \cdot H_\ell \cdot \left(\frac{1}{2}\right)^{H_{\ell-1}} \\
&= 0
\end{aligned}
\tag{61}
$$

**Variance** For the variance of the backpropagation term, we assume that $\frac{\partial \mathcal{L}}{\partial z_{h_\ell}}$ is uncorrelated with $\frac{\partial z_{h_\ell}}{\partial z_{h_{\ell-1}}}$.

$$
\begin{aligned}
Var\left[\frac{\partial \mathcal{L}}{\partial z_{h_{\ell-1}}}\right] &= H_\ell Var\left[\frac{\partial \mathcal{L}}{\partial z_{h_\ell}} \frac{\partial z_{h_\ell}}{\partial z_{h_{\ell-1}}}\right] \\
&\approx H_\ell \left(Var\left[\frac{\partial \mathcal{L}}{\partial z_{h_\ell}}\right] E\left[\frac{\partial z_{h_\ell}}{\partial z_{h_{\ell-1}}}\right]^2 + E\left[\frac{\partial \mathcal{L}}{\partial z_{h_\ell}}\right]^2 Var\left[\frac{\partial z_{h_\ell}}{\partial z_{h_{\ell-1}}}\right]\right. \\
&\left. + Var\left[\frac{\partial \mathcal{L}}{\partial z_{h_\ell}}\right] Var\left[\frac{\partial z_{h_\ell}}{\partial z_{h_{\ell-1}}}\right]\right)
\end{aligned}
\tag{62}
$$

Assuming again that $E\left[\frac{\partial \mathcal{L}}{\partial z_{h_\ell}}\right] = 0$, and reusing the result $E\left[\frac{\partial z_{h_\ell}}{\partial z_{h_{\ell-1}}}\right] = \left(\frac{1}{2}\right)^{H_{\ell-1}}$.

$$
Var\left[\frac{\partial \mathcal{L}}{\partial z_{h_{\ell-1}}}\right] \approx Var\left[\frac{\partial \mathcal{L}}{\partial z_{h_\ell}}\right] H_\ell \left(\left(\frac{1}{2}\right)^{2 \cdot H_{\ell-1}} + Var\left[\frac{\partial z_{h_\ell}}{\partial z_{h_{\ell-1}}}\right]\right)
\tag{63}
$$

Focusing now on $Var\left[\frac{\partial z_{h_\ell}}{\partial z_{h_{\ell-1}}}\right]$, we have:

$$
\begin{aligned}
Var\left[\frac{\partial z_{h_\ell}}{\partial z_{h_{\ell-1}}}\right] &= E\left[\left(\frac{z_{h_\ell}}{W_{h_{\ell-1},h_\ell} z_{h_{\ell-1}} + 1 - W_{h_{\ell-1},h_\ell}}\right)^2\right] E[W_{h_{\ell-1},h_\ell}^2] \\
&\quad - E\left[\frac{z_{h_\ell}}{W_{h_{\ell-1},h_\ell} z_{h_{\ell-1}} + 1 - W_{h_{\ell-1},h_\ell}}\right]^2 E[W_{h_{\ell-1},h_\ell}]^2
\end{aligned}
\tag{64}
$$

Inserting the result for the expectation $E\left[\frac{z_{h_\ell}}{W_{h_{\ell-1},h_\ell}z_{h_{\ell-1}}+1-W_{h_{\ell-1},h_\ell}}\right]$ and Initializing again $E[W_{h_{\ell-1},h_\ell}] = \frac{1}{2}$.

$$
\begin{aligned}
Var\left[\frac{\partial z_{h_\ell}}{\partial z_{h_{\ell-1}}}\right] &\approx E\left[\left(\frac{z_{h_\ell}}{W_{h_{\ell-1},h_\ell}z_{h_{\ell-1}}+1-W_{h_{\ell-1},h_\ell}}\right)^2\right]E[W^2_{h_{\ell-1},h_\ell}] \\
&\quad - \left(\frac{1}{2}\right)^{2\cdot(H_{\ell-1}-1)}\left(\frac{1}{2}\right)^2 \\
&= E\left[\left(\frac{z_{h_\ell}}{W_{h_{\ell-1},h_\ell}z_{h_{\ell-1}}+1-W_{h_{\ell-1},h_\ell}}\right)^2\right]E[W^2_{h_{\ell-1},h_\ell}] \\
&\quad - \left(\frac{1}{2}\right)^{2\cdot H_{\ell-1}}
\end{aligned}
\tag{65}
$$

Using the identity that $E[W^2_{h_{\ell-1},h_\ell}] = Var[W_{h_{\ell-1},h_\ell}] + E[W_{h_{\ell-1},h_\ell}]^2$, and again using $E[W_{h_{\ell-1},h_\ell}] = \frac{1}{2}$.

$$
\begin{aligned}
Var\left[\frac{\partial z_{h_\ell}}{\partial z_{h_{\ell-1}}}\right] &\approx E\left[\left(\frac{z_{h_\ell}}{W_{h_{\ell-1},h_\ell}z_{h_{\ell-1}}+1-W_{h_{\ell-1},h_\ell}}\right)^2\right]\left(Var[W_{h_{\ell-1},h_\ell}]+\frac{1}{4}\right) \\
&\quad - \left(\frac{1}{2}\right)^{2\cdot H_{\ell-1}}
\end{aligned}
\tag{66}
$$

To derive $E\left[\left(\frac{z_{h_\ell}}{W_{h_{\ell-1},h_\ell}z_{h_{\ell-1}}+1-W_{h_{\ell-1},h_\ell}}\right)^2\right]$ the result for $Var[z_{h_\ell}]$ can be used, but for $\hat{H}_{\ell-1} = H_{\ell-1} - 1$, because there is one less term. Inserting $E\left[\left(\frac{z_{h_\ell}}{W_{h_{\ell-1},h_\ell}z_{h_{\ell-1}}+1-W_{h_{\ell-1},h_\ell}}\right)^2\right] = \left(Var[W_{h_{\ell-1},h_\ell}]+\frac{1}{4}\right)^{H_{\ell-1}-1}\left(Var[z_{h_{\ell-1}}]+1\right)^{H_{\ell-1}-1}$, we have:

$$
\begin{aligned}
Var\left[\frac{\partial z_{h_\ell}}{\partial z_{h_{\ell-1}}}\right] &\approx \left(Var[W_{h_{\ell-1},h_\ell}]+\frac{1}{4}\right)^{H_{\ell-1}-1}\left(Var[z_{h_{\ell-1}}]+1\right)^{H_{\ell-1}-1} \\
&\quad \cdot\left(Var[W_{h_{\ell-1},h_\ell}]+\frac{1}{4}\right) - \left(\frac{1}{2}\right)^{2\cdot H_{\ell-1}} \\
&= \left(Var[W_{h_{\ell-1},h_\ell}]+\frac{1}{4}\right)^{H_{\ell-1}}\left(Var[z_{h_{\ell-1}}]+1\right)^{H_{\ell-1}-1} - \left(\frac{1}{2}\right)^{2\cdot H_{\ell-1}}
\end{aligned}
\tag{67}
$$

Inserting the result for $Var\left[\frac{\partial z_{h_\ell}}{\partial z_{h_{\ell-1}}}\right]$ into the result for $Var\left[\frac{\partial \mathcal{L}}{\partial z_{h_{\ell-1}}}\right]$:

$$
\begin{aligned}
Var\left[\frac{\partial \mathcal{L}}{\partial z_{h_{\ell-1}}}\right] &\approx Var\left[\frac{\partial \mathcal{L}}{\partial z_{h_\ell}}\right]H_\ell\left(\left(\frac{1}{2}\right)^{2\cdot H_{\ell-1}}\right. \\
&\quad \left. + \left(Var[W_{h_{\ell-1},h_\ell}]+\frac{1}{4}\right)^{H_{\ell-1}}\left(Var[z_{h_{\ell-1}}]+1\right)^{H_{\ell-1}-1} - \left(\frac{1}{2}\right)^{2\cdot H_{\ell-1}}\right) \\
&= Var\left[\frac{\partial \mathcal{L}}{\partial z_{h_\ell}}\right]H_\ell \\
&\quad \cdot\left(\left(Var[W_{h_{\ell-1},h_\ell}]+\frac{1}{4}\right)^{H_{\ell-1}}\left(Var[z_{h_{\ell-1}}]+1\right)^{H_{\ell-1}-1}\right)
\end{aligned}
\tag{68}
$$

### B.4.3 INITIALIZATION

The $W_{h_{\ell-1},h_\ell}$ should be initialized with $E[W_{h_{\ell-1},h_\ell}] = \frac{1}{2}$, in order to not bias towards inclusion or exclusion of $z_{h_{\ell-1}}$. Using the derived variance approximations (68), the variance should be according to the forward pass:

$$Var[W_{h_{\ell-1},h_\ell}] = \left((1 + Var[z_{h_\ell}])^{-H_{\ell-1}} Var[z_{h_\ell}] + (4 + 4Var[z_{h_\ell}])^{-H_{\ell-1}}\right)^{\frac{1}{H_{\ell-1}}} - \frac{1}{4} \quad (69)$$

And according to the backward pass it should be:

$$Var[W_{h_{\ell-1},h_\ell}] = \left(\frac{(Var[z_{h_\ell}] + 1)^{1-H_{\ell-1}}}{H_\ell}\right)^{\frac{1}{H_{\ell-1}}} - \frac{1}{4} \quad (70)$$

Both criteria are dependent on the input variance. If the input variance is know then optimal initialization is possible. However, as this is often not the case one can perhaps assume that $Var[z_{h_{\ell-1}}] = 1$. This is not an unreasonable assumption in many cases, as there may either be a normalization layer somewhere or the input is normalized. If unit variance is assumed, the variance for the forward pass becomes:

$$Var[W_{h_{\ell-1},h_\ell}] = \left(2^{-H_{\ell-1}} + 8^{-H_{\ell-1}}\right)^{\frac{1}{H_{\ell-1}}} - \frac{1}{4} = \frac{1}{8}\left(\left(4^{H_{\ell-1}} + 1\right)^{H_{\ell-1}} - 2\right) \quad (71)$$

And from the backward pass:

$$Var[W_{h_{\ell-1},h_\ell}] = \left(\frac{2^{1-H_{\ell-1}}}{H_\ell}\right)^{\frac{1}{H_{\ell-1}}} - \frac{1}{4} \quad (72)$$

The variance requirement for both the forward and backward pass can be satisfied with $Var[W_{h_{\ell-1},h_\ell}] = \frac{1}{4}$ for a large $H_{\ell-1}$.

## C ARITHMETIC TASK

The aim of the "Arithmetic task" is to directly test arithmetic models ability to extrapolate beyond the training range. Additionally, our generalized version provides a high degree of flexibility in how the input is shaped, sampled, and the problem complexity.

Our "arithmetic task" is identical to the "simple function task" in the NALU paper (Trask et al., 2018). However, as they do not describe their setup in details, we use the setup from Madsen & Johansen (2019), which provide Algorithm 3, an evaluation-criterion to if and when the model has converged, the sparsity error, as well as methods for computing confidence intervals for success-rate and the sparsity error.

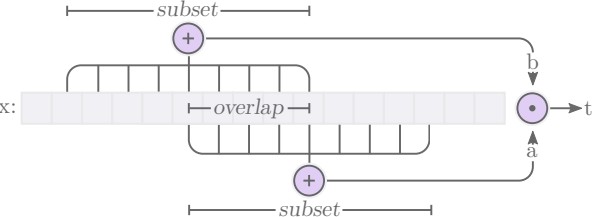

Figure 6: Shows how the dataset is parameterized.

### C.1 DATASET GENERATION

The goal is to sum two random subsets of a vector $\mathbf{x}$ ($a$ and $b$), and perform an arithmetic operation on these ($a \circ b$).

$$a = \sum_{i=s_{1,\text{start}}}^{s_{1,\text{end}}} x_i, \quad b = \sum_{i=s_{2,\text{start}}}^{s_{2,\text{end}}} x_i, \quad t = a \circ b \tag{73}$$

Algorithm 1 defines the exact procedure to generate the data, where an interpolation range will be used for training and validation and an extrapolation range will be used for testing. Default values are defined in table 3.

Table 3: Default dataset parameters for "Arithmetic task"

| Parameter name | Default value | Parameter name | Default value |
|---|---|---|---|
| Input size | 100 | Interpolation range | $U[1, 2]$ |
| Subset ratio | 0.25 | Extrapolation range | $U[2, 6]$ |
| Overlap ratio | 0.5 | | |

---

**Algorithm 1** Dataset generation algorithm for "Arithmetic task"

---

1: **function** DATASET(OP$(\cdot, \cdot)$ : Operation, $i$ : InputSize, $s$ : SubsetRatio, $o$ : OverlapRatio, $R$ : Range)
2:      $\mathbf{x} \leftarrow$ UNIFORM$(R_{lower}, R_{upper}, i)$        ▷ Sample $i$ elements uniformly
3:      $k \leftarrow$ UNIFORM$(0, 1 - 2s - o)$        ▷ Sample offset
4:      $a \leftarrow$ SUM$(\mathbf{x}[ik : i(k + s)])$        ▷ Create sum $a$ from subset
5:      $b \leftarrow$ SUM$(\mathbf{x}[i(k + s - o) : i(k + 2s - 0)])$        ▷ Create sum $b$ from subset
6:      $t \leftarrow$ OP$(a, b)$        ▷ Perform operation on $a$ and $b$
7:      **return** $x, t$

---

### C.2 MODEL DEFINTIONS AND SETUP

Models are defined in table 4 and are all optimized with Adam optimization (Kingma & Ba, 2014) using default parameters, and trained over $5 \cdot 10^6$ iterations. Training takes about 8 hours on a single CPU core(8-Core Intel Xeon E5-2665 2.4GHz). We run 19150 experiments on a HPC cluster.

The training dataset is continuously sampled from the interpolation range where a different seed is used for each experiment, all experiments use a mini-batch size of 128 observations, a fixed validation dataset with $1 \cdot 10^4$ observations sampled from the interpolation range, and a fixed test dataset with $1 \cdot 10^4$ observations sampled from the extrapolation range.

Table 4: Model definitions

| Model | Layer 1 | Layer 2 | $\hat{\lambda}_{\text{sparse}}$ | $\lambda_{\text{start}}$ | $\lambda_{\text{end}}$ |
|---|---|---|---|---|---|
| NMU | NAU | NMU | 10 | $10^6$ | $2 \cdot 10^6$ |
| NAU | NAU | NAU | 0.01 | $5 \cdot 10^3$ | $5 \cdot 10^4$ |
| NAC$_\bullet$ | NAC$_+$ | NAC$_\bullet$ | – | – | – |
| NAC$_{\bullet,\sigma}$ | NAC$_+$ | NAC$_{\bullet,\sigma}$ | – | – | – |
| NAC$_{\bullet,\text{NMU}}$ | NAC$_+$ | NAC$_{\bullet,\text{NMU}}$ | 10 | $10^6$ | $2 \cdot 10^6$ |
| NAC$_+$ | NAC$_+$ | NAC$_+$ | – | – | – |
| NALU | NALU | NALU | – | – | – |
| Linear | Linear | Linear | – | – | – |
| ReLU | ReLU | ReLU | – | – | – |
| ReLU6 | ReLU6 | ReLU6 | – | – | – |

### C.3 ABLATION STUDY

To validate our model, we perform an ablation on the multiplication problem. Some noteworthy observations:

1. None of the $W$ constraints, such as $\mathcal{R}_{sparse}$ and clamping W to be in $[0, 1]$, are necessary when the hidden size is just 2.

2. Removing the $\mathcal{R}_{sparse}$ causes the NMU to immediately fail for larger hidden sizes.

3. Removing the clamping of W does not cause much difference. This is because $\mathcal{R}_{sparse}$ also constrains $W$ outside of $[0, 1]$. The regularizer used here is $\mathcal{R}_{sparse} = \min(|W|, |1 - W|)$, which is identical to the one used in other experiments in $[0, 1]$, but is also valid outside $[0, 1]$. Doing this gives only a slightly slower convergence. Although, this can not be guaranteed in general, as the regularizer is omitted during the initial optimization.

4. Removing both constraints, gives a somewhat satisfying solution, but with a lower success-rate, slower convergence, and higher sparsity error.

In conclusion both constraints are valuable, as they provide faster convergence and a sparser solution, but they are not critical to the success-rate of the NMU.

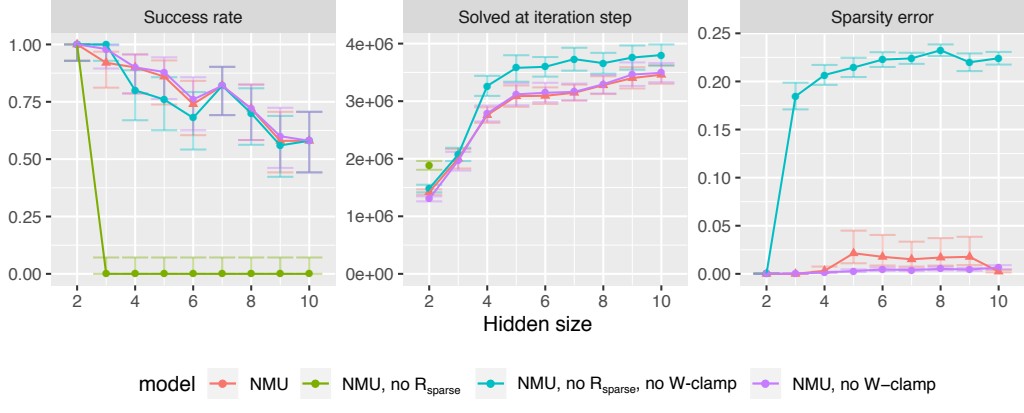

Figure 7: Ablation study where $\mathcal{R}_{sparse}$ is removed and the clamping of W is removed. There are 50 experiments with different seeds, for each configuration.

## C.4 EFFECT OF DATASET PARAMETER

To stress test the models on the multiplication task, we vary the dataset parameters one at a time while keeping the others at their default value (default values in table 3). Each runs for 50 experiments with different seeds. The results, are visualized in figure 8.

In figure 3, the interpolation-range is changed, therefore the extrapolation-range needs to be changed such it doesn't overlap. For each interpolation-range the following extrapolation-range is used: $U[-2, -1]$ uses $U[-6, -2]$, $U[-2, 2]$ uses $U[-6, -2] \cup U[2, 6]$, $U[0, 1]$ uses $U[1, 5]$, $U[0.1, 0.2]$ uses $U[0.2, 2]$, $U[1.1, 1.2]$ uses $U[1.2, 6]$, $U[1, 2]$ uses $U[2, 6]$, $U[10, 20]$ uses $U[20, 40]$.

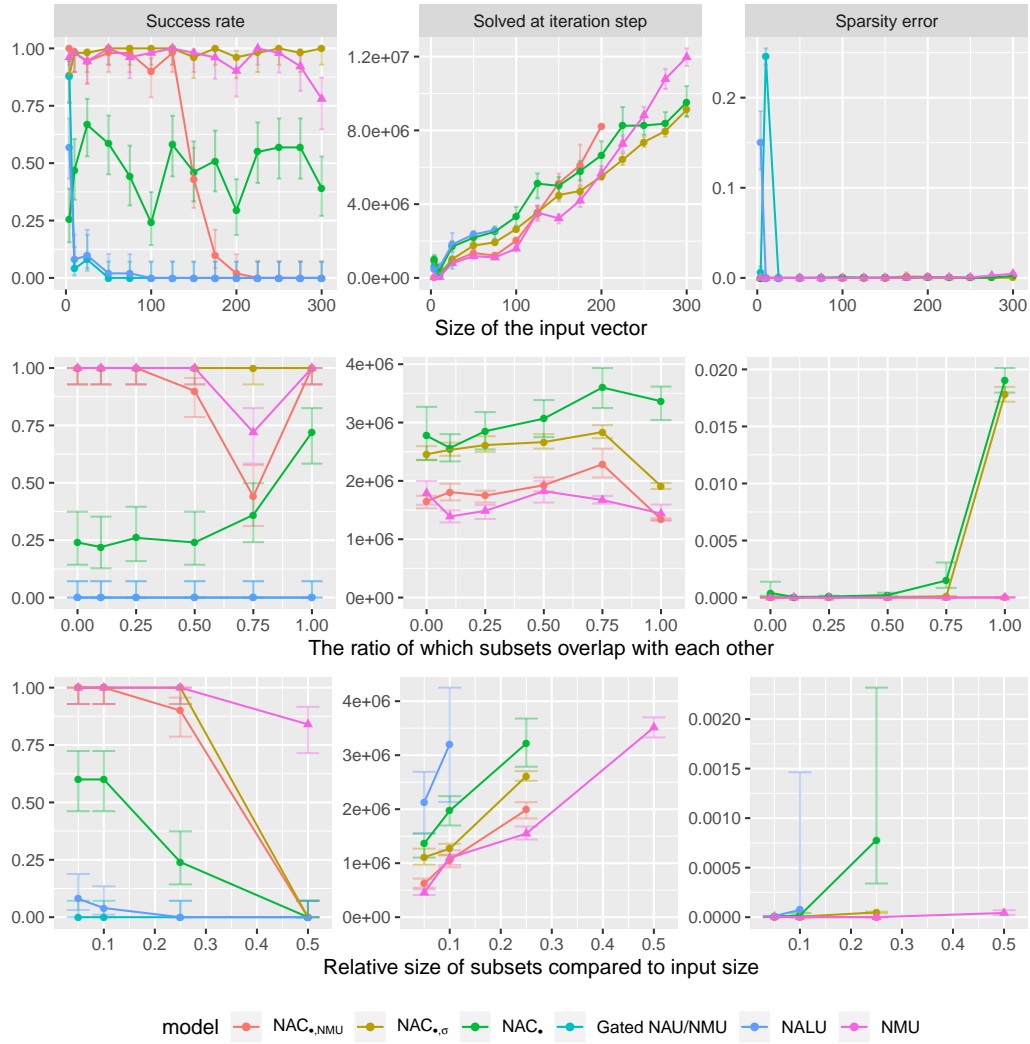

Figure 8: Shows the effect of the dataset parameters.

## C.5 GATING CONVERGENCE EXPERIMENT

In the interest of adding some understand of what goes wrong in the NALU gate, and the shared weight choice that NALU employs to remedy this, we introduce the following experiment.

We train two models to fit the arithmetic task. Both uses the $NAC_{+}$ in the first layer and NALU in the second layer. The only difference is that one model shares the weight between $NAC_{+}$ and $NAC_{\bullet}$ in the NALU, and the other treat them as two separate units with separate weights. In both cases NALU should gate between $NAC_{+}$ and $NAC_{\bullet}$ and choose the appropriate operation. Note that this NALU

model is different from the one presented elsewhere in this paper, including the original NALU paper (Trask et al., 2018). The typical NALU model is just two NALU layers with shared weights.

Furthermore, we also introduce a new gated unit that simply gates between our proposed NMU and NAU, using the same sigmoid gating-mechanism as in the NALU. This combination is done with seperate weights, as NMU and NAU use different weight constrains and can therefore not be shared.

The models are trained and evaluated over 100 different seeds on the multiplication and addition task. A histogram of the gate-value for all seeds is presented in figure 9 and table 5 contains a summary. Some noteworthy observations:

1. When the NALU weights are separated far more trials converge to select $NAC_+$ for both the addition and multiplication task. Sharing the weights between $NAC_+$ and $NAC_\bullet$ makes the gating less likely to converge for addition.

2. The performance of the addition task is dependent on NALU selecting the right operation. In the multiplication task, when the right gate is selected, $NAC_\bullet$ do not converge consistently, unlike our NMU that converges more consistently.

3. Which operation the gate converges to appears to be mostly random and independent of the task. These issues are caused by the sigmoid gating-mechanism and thus exists independent of the used sub-units.

These observations validates that the NALU gating-mechanism does not converge as intended. This becomes a critical issues when more gates are present, as is normally the case. E.g. when stacking multiple NALU layers together.

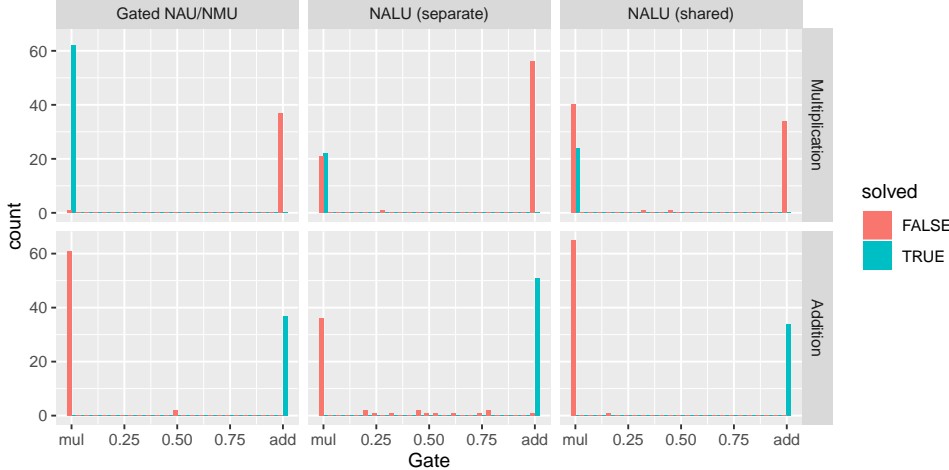

Figure 9: Shows the gating-value in the NALU layer and a variant that uses NAU/NMU instead of $NAC_+$/$NAC_\bullet$. Separate/shared refers to the weights in $NAC_+$/$NAC_\bullet$ used in NALU.

Table 5: Comparison of the success-rate, when the model converged, and the sparsity error, with 95% confidence interval on the "arithmetic datasets" task. Each value is a summary of 100 different seeds.

| Op | Model | Success | Solved at iteration step | | Sparsity error |
|---|---|---|---|---|---|
| | | Rate | Median | Mean | Mean |
| $\times$ | Gated NAU/NMU | $\mathbf{62}\%\ ^{+9\%}_{-10\%}$ | $\mathbf{1.5 \cdot 10^6}$ | $\mathbf{1.5 \cdot 10^6}\ ^{+3.9\cdot10^4}_{-3.8\cdot10^4}$ | $\mathbf{5.0 \cdot 10^{-5}}\ ^{+2.3\cdot10^{-5}}_{-1.8\cdot10^{-5}}$ |
| | NALU (separate) | $22\%\ ^{+9\%}_{-7\%}$ | $2.8 \cdot 10^6$ | $3.3 \cdot 10^6\ ^{+3.9\cdot10^5}_{-3.6\cdot10^5}$ | $5.8 \cdot 10^{-2}\ ^{+4.1\cdot10^{-2}}_{-2.3\cdot10^{-2}}$ |
| | NALU (shared) | $24\%\ ^{+9\%}_{-7\%}$ | $2.9 \cdot 10^6$ | $3.3 \cdot 10^6\ ^{+3.7\cdot10^5}_{-3.6\cdot10^5}$ | $1.0 \cdot 10^{-3}\ ^{+1.1\cdot10^{-3}}_{-4.5\cdot10^{-4}}$ |
| $+$ | Gated NAU/NMU | $37\%\ ^{+10\%}_{-9\%}$ | $\mathbf{1.9 \cdot 10^4}$ | $4.2 \cdot 10^5\ ^{+7.3\cdot10^4}_{-6.7\cdot10^4}$ | $\mathbf{1.7 \cdot 10^{-1}}\ ^{+4.6\cdot10^{-2}}_{-4.0\cdot10^{-2}}$ |
| | NALU (separate) | $\mathbf{51}\%\ ^{+10\%}_{-10\%}$ | $1.4 \cdot 10^5$ | $\mathbf{2.9 \cdot 10^5}\ ^{+3.5\cdot10^4}_{-4.3\cdot10^4}$ | $1.8 \cdot 10^{-1}\ ^{+1.4\cdot10^{-2}}_{-1.4\cdot10^{-2}}$ |
| | NALU (shared) | $34\%\ ^{+10\%}_{-9\%}$ | $1.8 \cdot 10^5$ | $3.1 \cdot 10^5\ ^{+4.3\cdot10^4}_{-5.4\cdot10^4}$ | $1.8 \cdot 10^{-1}\ ^{+2.3\cdot10^{-2}}_{-2.1\cdot10^{-2}}$ |

## C.6 Regularization

The $\lambda_{start}$ and $\lambda_{end}$ are simply selected based on how much time it takes for the model to converge. The sparsity regularizer should not be used during early optimization as this part of the optimization is exploratory and concerns finding the right solution by getting each weight on the right side of $\pm 0.5$.

In figure 10, 11 and 12 the scaling factor $\hat{\lambda}_{\text{sparse}}$ is optimized.

$$\lambda_{\text{sparse}} = \hat{\lambda}_{\text{sparse}} \max(\min(\frac{t - \lambda_{\text{start}}}{\lambda_{\text{end}} - \lambda_{\text{start}}}, 1), 0) \tag{74}$$

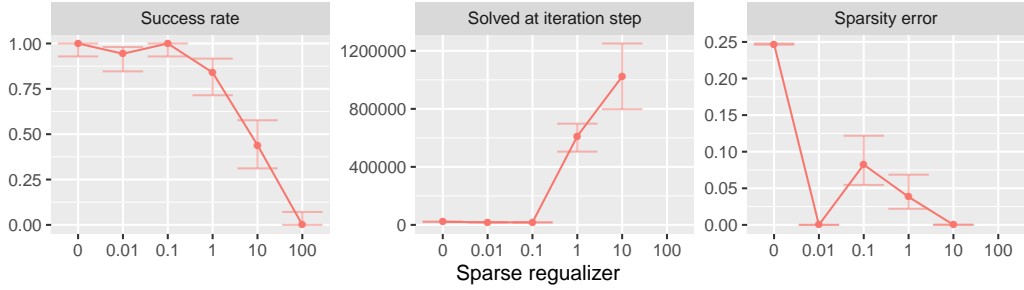

Figure 10: Shows effect of $\hat{\lambda}_{\text{sparse}}$ in NAU on the arithmetic dataset for the $+$ operation.

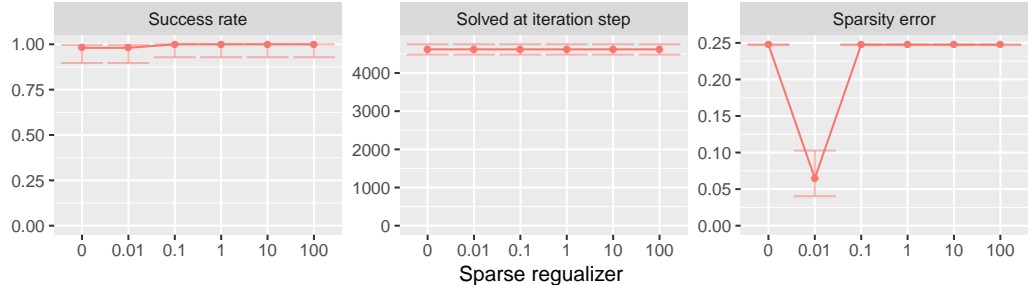

Figure 11: Shows effect of $\hat{\lambda}_{\text{sparse}}$ in NAU on the arithmetic dataset for the $-$ operation.

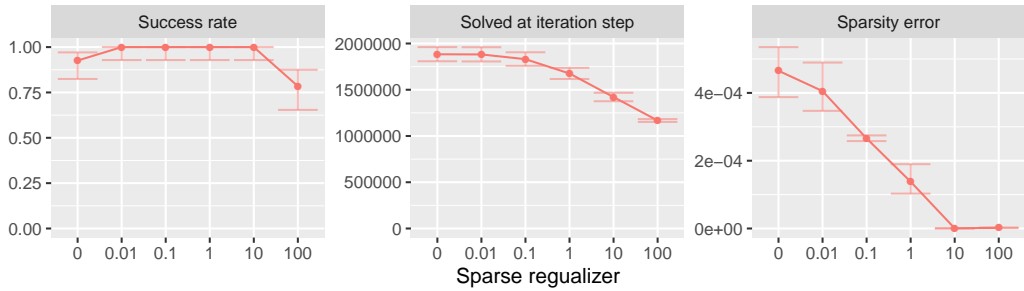

Figure 12: Shows effect of $\hat{\lambda}_{\text{sparse}}$ in NMU on the arithmetic dataset for the $\times$ operation.

## C.7 Comparing all models

Table 6 compares all models on all operations used in NALU (Trask et al., 2018). All variations of models and operations are trained for 100 different seeds to build confidence intervals. Some noteworthy observations are:

1. Division does not work for any model, including the $NAC_\bullet$ and NALU models. This may seem surprising but is actually in line with the results from the NALU paper (Trask et al. (2018), table 1) where there is a large error given the interpolation range. The extrapolation range has a smaller error, but this is an artifact of their evaluation method where they normalize with a random baseline. Since a random baseline will have a higher error for the extrapolation range, errors just appear to be smaller. A correct solution to division should have both a small interpolation and extrapolation error.

2. $NAC_\bullet$ and NALU are barely able to learn $\sqrt{z}$, with just 2% success-rate for NALU and 7% success-rate for $NAC_\bullet$.

3. NMU is fully capable of learning $z^2$. It learn this by learning the same subset twice in the NAU layer, this is also how $NAC_\bullet$ learn $z^2$.

4. The Gated NAU/NMU (discussed in section C.5) works very poorly, because the NMU initialization assumes that $E[z_{h_{\ell-1}}] = 0$. This is usually true, as discussed in section 2.6, but not in this case for the first layer. In the recommended NMU model, the NMU layer appears after NAU, which causes that assumption to be satisfied.

Table 6: Comparison of the success-rate, when the model converged, and the sparsity error, with 95% confidence interval on the "arithmetic datasets" task. Each value is a summary of 100 different seeds.

| Op | Model | Success Rate | Solved at iteration step | | Sparsity error Mean |
|---|---|---|---|---|---|
| | | | Median | Mean | |
| $\times$ | $NAC_{\bullet,NMU}$ | $93\%\,^{+4\%}_{-7\%}$ | $1.8 \cdot 10^6$ | $2.0 \cdot 10^6\,^{+1.0\cdot10^5}_{-9.7\cdot10^4}$ | $9.5 \cdot 10^{-7}\,^{+4.2\cdot10^{-7}}_{-4.2\cdot10^{-7}}$ |
| | $NAC_{\bullet,\sigma}$ | $\mathbf{100\%}\,^{+0\%}_{-4\%}$ | $2.5 \cdot 10^6$ | $2.6 \cdot 10^6\,^{+8.8\cdot10^4}_{-7.2\cdot10^4}$ | $4.6 \cdot 10^{-5}\,^{+5.0\cdot10^{-6}}_{-5.6\cdot10^{-6}}$ |
| | $NAC_\bullet$ | $31\%\,^{+10\%}_{-8\%}$ | $2.8 \cdot 10^6$ | $3.0 \cdot 10^6\,^{+2.9\cdot10^5}_{-2.4\cdot10^5}$ | $5.8 \cdot 10^{-4}\,^{+4.8\cdot10^{-4}}_{-2.6\cdot10^{-4}}$ |
| | $NAC_+$ | $0\%\,^{+4\%}_{-0\%}$ | — | — | — |
| | Gated $^{NAU}_{NMU}$ | $0\%\,^{+4\%}_{-0\%}$ | — | — | — |
| | Linear | $0\%\,^{+4\%}_{-0\%}$ | — | — | — |
| | NALU | $0\%\,^{+4\%}_{-0\%}$ | — | — | — |
| | NAU | $0\%\,^{+4\%}_{-0\%}$ | — | — | — |
| | NMU | $\mathbf{98\%}\,^{+1\%}_{-5\%}$ | $\mathbf{1.4 \cdot 10^6}$ | $\mathbf{1.5 \cdot 10^6}\,^{+5.0\cdot10^4}_{-6.6\cdot10^4}$ | $\mathbf{4.2 \cdot 10^{-7}}\,^{+2.9\cdot10^{-8}}_{-2.9\cdot10^{-8}}$ |
| | ReLU | $0\%\,^{+4\%}_{-0\%}$ | — | — | — |
| | ReLU6 | $0\%\,^{+4\%}_{-0\%}$ | — | — | — |
| / | $NAC_{\bullet,NMU}$ | $\mathbf{0\%}\,^{+4\%}_{-0\%}$ | — | — | — |
| | $NAC_{\bullet,\sigma}$ | $\mathbf{0\%}\,^{+4\%}_{-0\%}$ | — | — | — |
| | $NAC_\bullet$ | $\mathbf{0\%}\,^{+4\%}_{-0\%}$ | — | — | — |
| | $NAC_+$ | $\mathbf{0\%}\,^{+4\%}_{-0\%}$ | — | — | — |
| | Gated $^{NAU}_{NMU}$ | $\mathbf{0\%}\,^{+4\%}_{-0\%}$ | — | — | — |
| | Linear | $\mathbf{0\%}\,^{+4\%}_{-0\%}$ | — | — | — |
| | NALU | $\mathbf{0\%}\,^{+4\%}_{-0\%}$ | — | — | — |
| | NAU | $\mathbf{0\%}\,^{+4\%}_{-0\%}$ | — | — | — |
| | NMU | $\mathbf{0\%}\,^{+4\%}_{-0\%}$ | — | — | — |
| | ReLU | $\mathbf{0\%}\,^{+4\%}_{-0\%}$ | — | — | — |
| | ReLU6 | $\mathbf{0\%}\,^{+4\%}_{-0\%}$ | — | — | — |

Table 6: Comparison of the success-rate, when the model converged, and the sparsity error, with 95% confidence interval on the "arithmetic datasets" task. Each value is a summary of 100 different seeds. *(continued)*

| Op | Model | Success | Solved at | | Sparsity error |
|---|---|---|---|---|---|
| | | Rate | Median | Mean | Mean |
| + | $\mathrm{NAC}_{\bullet,\mathrm{NMU}}$ | $0\%\ ^{+4\%}_{-0\%}$ | — | — | — |
| | $\mathrm{NAC}_{\bullet,\sigma}$ | $0\%\ ^{+4\%}_{-0\%}$ | — | — | — |
| | $\mathrm{NAC}_{\bullet}$ | $0\%\ ^{+4\%}_{-0\%}$ | — | — | — |
| | $\mathrm{NAC}_{+}$ | $\mathbf{100\%}\ ^{+0\%}_{-4\%}$ | $2.5 \cdot 10^5$ | $4.9 \cdot 10^5\ ^{+5.2\cdot10^4}_{-4.5\cdot10^4}$ | $2.3 \cdot 10^{-1}\ ^{+6.5\cdot10^{-3}}_{-6.5\cdot10^{-3}}$ |
| | $\mathrm{Gated}^{\mathrm{NAU}}_{\mathrm{NMU}}$ | $0\%\ ^{+4\%}_{-0\%}$ | — | — | — |
| | Linear | $\mathbf{100\%}\ ^{+0\%}_{-4\%}$ | $6.1 \cdot 10^4$ | $\mathbf{6.3 \cdot 10^4}\ ^{+2.5\cdot10^3}_{-3.3\cdot10^3}$ | $2.5 \cdot 10^{-1}\ ^{+3.6\cdot10^{-4}}_{-3.6\cdot10^{-4}}$ |
| | NALU | $14\%\ ^{+8\%}_{-5\%}$ | $1.5 \cdot 10^6$ | $1.6 \cdot 10^6\ ^{+3.8\cdot10^5}_{-3.3\cdot10^5}$ | $1.7 \cdot 10^{-1}\ ^{+2.7\cdot10^{-2}}_{-2.5\cdot10^{-2}}$ |
| | NAU | $\mathbf{100\%}\ ^{+0\%}_{-4\%}$ | $\mathbf{1.8 \cdot 10^4}$ | $3.9 \cdot 10^5\ ^{+4.5\cdot10^4}_{-3.7\cdot10^4}$ | $\mathbf{3.2 \cdot 10^{-5}}\ ^{+1.3\cdot10^{-5}}_{-1.3\cdot10^{-5}}$ |
| | NMU | $0\%\ ^{+4\%}_{-0\%}$ | — | — | — |
| | ReLU | $62\%\ ^{+9\%}_{-10\%}$ | $6.2 \cdot 10^4$ | $7.6 \cdot 10^4\ ^{+8.3\cdot10^3}_{-7.0\cdot10^3}$ | $2.5 \cdot 10^{-1}\ ^{+2.4\cdot10^{-3}}_{-2.4\cdot10^{-3}}$ |
| | ReLU6 | $0\%\ ^{+4\%}_{-0\%}$ | — | — | — |
| − | $\mathrm{NAC}_{\bullet,\mathrm{NMU}}$ | $0\%\ ^{+4\%}_{-0\%}$ | — | — | — |
| | $\mathrm{NAC}_{\bullet,\sigma}$ | $0\%\ ^{+4\%}_{-0\%}$ | — | — | — |
| | $\mathrm{NAC}_{\bullet}$ | $0\%\ ^{+4\%}_{-0\%}$ | — | — | — |
| | $\mathrm{NAC}_{+}$ | $\mathbf{100\%}\ ^{+0\%}_{-4\%}$ | $9.0 \cdot 10^3$ | $3.7 \cdot 10^5\ ^{+3.8\cdot10^4}_{-3.8\cdot10^4}$ | $2.3 \cdot 10^{-1}\ ^{+5.4\cdot10^{-3}}_{-5.4\cdot10^{-3}}$ |
| | $\mathrm{Gated}^{\mathrm{NAU}}_{\mathrm{NMU}}$ | $0\%\ ^{+4\%}_{-0\%}$ | — | — | — |
| | Linear | $7\%\ ^{+7\%}_{-4\%}$ | $3.3 \cdot 10^6$ | $1.4 \cdot 10^6\ ^{+7.0\cdot10^5}_{-6.1\cdot10^5}$ | $1.8 \cdot 10^{-1}\ ^{+7.2\cdot10^{-2}}_{-5.8\cdot10^{-2}}$ |
| | NALU | $14\%\ ^{+8\%}_{-5\%}$ | $1.9 \cdot 10^6$ | $1.9 \cdot 10^6\ ^{+4.4\cdot10^5}_{-4.5\cdot10^5}$ | $2.1 \cdot 10^{-1}\ ^{+2.2\cdot10^{-2}}_{-2.2\cdot10^{-2}}$ |
| | NAU | $\mathbf{100\%}\ ^{+0\%}_{-4\%}$ | $\mathbf{5.0 \cdot 10^3}$ | $\mathbf{1.6 \cdot 10^5}\ ^{+1.7\cdot10^4}_{-1.6\cdot10^4}$ | $6.6 \cdot 10^{-2}\ ^{+2.5\cdot10^{-2}}_{-1.9\cdot10^{-2}}$ |
| | NMU | $56\%\ ^{+9\%}_{-10\%}$ | $1.0 \cdot 10^6$ | $1.0 \cdot 10^6\ ^{+5.8\cdot10^2}_{-5.8\cdot10^2}$ | $\mathbf{3.4 \cdot 10^{-4}}\ ^{+3.2\cdot10^{-5}}_{-2.6\cdot10^{-5}}$ |
| | ReLU | $0\%\ ^{+4\%}_{-0\%}$ | — | — | — |
| | ReLU6 | $0\%\ ^{+4\%}_{-0\%}$ | — | — | — |
| $\sqrt{z}$ | $\mathrm{NAC}_{\bullet,\mathrm{NMU}}$ | $3\%\ ^{+5\%}_{-2\%}$ | $1.0 \cdot 10^6$ | $\mathbf{1.0 \cdot 10^6}\ ^{+NaN\cdot10^{-Inf}}_{-NaN\cdot10^{-Inf}}$ | $\mathbf{1.7 \cdot 10^{-1}}\ ^{+8.3\cdot10^{-3}}_{-8.1\cdot10^{-3}}$ |
| | $\mathrm{NAC}_{\bullet,\sigma}$ | $0\%\ ^{+4\%}_{-0\%}$ | — | — | — |
| | $\mathrm{NAC}_{\bullet}$ | $7\%\ ^{+7\%}_{-4\%}$ | $\mathbf{4.0 \cdot 10^5}$ | $1.5 \cdot 10^6\ ^{+6.0\cdot10^5}_{-5.6\cdot10^5}$ | $2.4 \cdot 10^{-1}\ ^{+1.7\cdot10^{-2}}_{-1.7\cdot10^{-2}}$ |
| | $\mathrm{NAC}_{+}$ | $0\%\ ^{+4\%}_{-0\%}$ | — | — | — |
| | $\mathrm{Gated}^{\mathrm{NAU}}_{\mathrm{NMU}}$ | $0\%\ ^{+4\%}_{-0\%}$ | — | — | — |
| | Linear | $0\%\ ^{+4\%}_{-0\%}$ | — | — | — |
| | NALU | $2\%\ ^{+5\%}_{-1\%}$ | $2.6 \cdot 10^6$ | $3.3 \cdot 10^6\ ^{+1.8\cdot10^6}_{-2.2\cdot10^6}$ | $5.0 \cdot 10^{-1}\ ^{+2.5\cdot10^{-6}}_{-8.0\cdot10^{-6}}$ |
| | NAU | $0\%\ ^{+4\%}_{-0\%}$ | — | — | — |
| | NMU | $0\%\ ^{+4\%}_{-0\%}$ | — | — | — |
| | ReLU | $0\%\ ^{+4\%}_{-0\%}$ | — | — | — |
| | ReLU6 | $0\%\ ^{+4\%}_{-0\%}$ | — | — | — |

Table 6: Comparison of the success-rate, when the model converged, and the sparsity error, with 95% confidence interval on the "arithmetic datasets" task. Each value is a summary of 100 different seeds. *(continued)*

| Op | Model | Success | Solved at | | Sparsity error |
|---|---|---|---|---|---|
| | | Rate | Median | Mean | Mean |
| | $\text{NAC}_{\bullet,\text{NMU}}$ | $\mathbf{100\%} \, ^{+0\%}_{-4\%}$ | $1.4 \cdot 10^6$ | $1.5 \cdot 10^6 \, ^{+8.4 \cdot 10^4}_{-7.9 \cdot 10^4}$ | $\mathbf{2.9 \cdot 10^{-7}} \, ^{+1.4 \cdot 10^{-8}}_{-1.4 \cdot 10^{-8}}$ |
| | $\text{NAC}_{\bullet,\sigma}$ | $\mathbf{100\%} \, ^{+0\%}_{-4\%}$ | $1.9 \cdot 10^6$ | $1.9 \cdot 10^6 \, ^{+5.3 \cdot 10^4}_{-6.2 \cdot 10^4}$ | $1.8 \cdot 10^{-2} \, ^{+4.3 \cdot 10^{-4}}_{-4.3 \cdot 10^{-4}}$ |
| | $\text{NAC}_{\bullet}$ | $77\% \, ^{+7\%}_{-9\%}$ | $3.3 \cdot 10^6$ | $3.2 \cdot 10^6 \, ^{+1.6 \cdot 10^5}_{-2.0 \cdot 10^5}$ | $1.8 \cdot 10^{-2} \, ^{+5.8 \cdot 10^{-4}}_{-5.7 \cdot 10^{-4}}$ |
| | $\text{NAC}_{+}$ | $0\% \, ^{+4\%}_{-0\%}$ | — | — | — |
| | $\text{Gated}^{\text{NAU}}_{\text{NMU}}$ | $0\% \, ^{+4\%}_{-0\%}$ | — | — | — |
| | Linear | $0\% \, ^{+4\%}_{-0\%}$ | — | — | — |
| $z^2$ | NALU | $0\% \, ^{+4\%}_{-0\%}$ | — | — | — |
| | NAU | $0\% \, ^{+4\%}_{-0\%}$ | — | — | — |
| | NMU | $\mathbf{100\%} \, ^{+0\%}_{-4\%}$ | $\mathbf{1.2 \cdot 10^6}$ | $\mathbf{1.3 \cdot 10^6} \, ^{+3.1 \cdot 10^4}_{-3.6 \cdot 10^4}$ | $3.7 \cdot 10^{-5} \, ^{+5.4 \cdot 10^{-5}}_{-3.7 \cdot 10^{-5}}$ |
| | ReLU | $0\% \, ^{+4\%}_{-0\%}$ | — | — | — |
| | ReLU6 | $0\% \, ^{+4\%}_{-0\%}$ | — | — | — |

# D  SEQUENTIAL MNIST

## D.1  TASK AND EVALUATION CRITERIA

The simple function task is a purely synthetic task, that does not require a deep network. As such it does not test if an arithmetic layer inhibits the networks ability to be optimized using gradient decent.

The sequential MNIST task takes the numerical value of a sequence of MNIST digits and applies a binary operation recursively. Such that $t_i = Op(t_{i-1}, z_t)$, where $z_t$ is the MNIST digit's numerical value. This is identical to the "MNIST Counting and Arithmetic Tasks" in Trask et al. (2018, section 4.2). We present the addition variant to validate the NAU's ability to backpropagate, and we add an additional multiplication variant to validate the NMU's ability to backpropagate.

The performance of this task depends on the quality of the image-to-scalar network and the arithmetic layer's ability to model the scalar. We use mean-square-error (MSE) to evaluate joint image-to-scalar and arithmetic layer model performance. To determine an MSE threshold from the correct prediction we use an empirical baseline. This is done by letting the arithmetic layer be solved, such that only the image-to-scalar is learned. By learning this over multiple seeds an upper bound for an MSE threshold can be set. In our experiment we use the 1% one-sided upper confidence-interval, assuming a student-t distribution.

Similar to the simple function task we use a success-criteria as reporting the MSE is not interpretable and models that do not converge will obscure the mean. Furthermore, because the operation is applied recursively, natural error from the dataset will accumulate over time, thus exponentially increasing the MSE. Using a baseline model and reporting the successfulness solves this interpretation challenge.

## D.2  ADDITION OF SEQUENTIAL MNIST

Figure 13 shows results for sequential addition of MNIST digits. This experiment is identical to the MNIST Digit Addition Test from Trask et al. (2018, section 4.2). The models are trained on a sequence of 10 digits and evaluated on sequences between 1 and 1000 MNIST digits.

Note that the NAU model includes the $R_z$ regularizer, similarly to the "Multiplication of sequential MNIST" experiment in section 4.2. However, because the weights are in $[-1, 1]$, and not $[0, 1]$, and the idendity of addition is 0, and not 1, $R_z$ is

$$\mathcal{R}_z = \frac{1}{H_{\ell-1}H_\ell} \sum_{h_\ell}^{H_\ell} \sum_{h_{\ell-1}}^{H_{\ell-1}} (1 - |W_{h_{\ell-1},h_\ell}|) \cdot \bar{z}_{h_{\ell-1}}^2 \,. \tag{75}$$

To provide a fair comparison, a variant of $\mathrm{NAC}_+$ that also uses this regularizer is included, this variant is called $\mathrm{NAC}_{+,R_z}$. Section D.3 provides an ablation study of the $R_z$ regularizer.

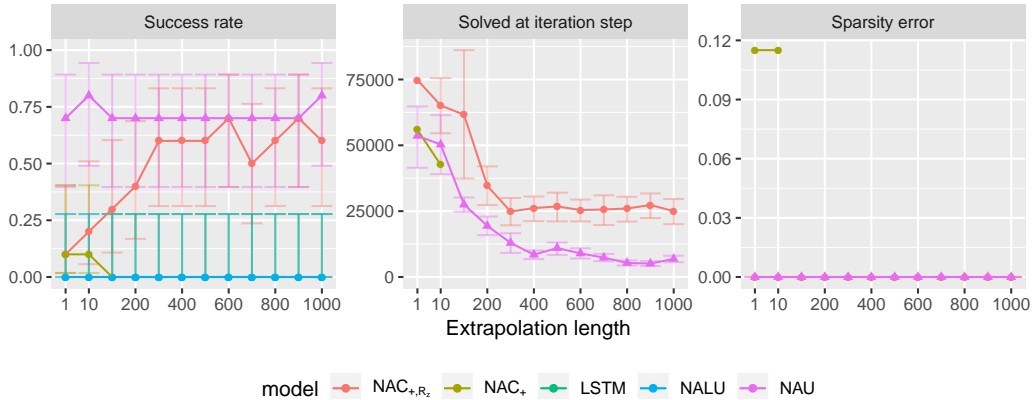

Figure 13: Shows the ability of each model to learn the arithmetic operation of addition and backpropagate through the arithmetic layer in order to learn an image-to-scalar value for MNIST digits. The model is tested by extrapolating to larger sequence lengths than what it has been trained on. The NAU and $\mathrm{NAC}_{+,R_z}$ models use the $\mathrm{R}_z$ regularizer from section 4.2.

## D.3 SEQUENTIAL ADDTION WITHOUT THE $R_z$ REGULARIZER

As an ablation study of the $R_z$ regularizer, figure 14 shows the NAU model without the $R_z$ regularizer. Removing the regularizer causes a reduction in the success-rate. The reduction is likely larger, as compared to sequential multiplication, because the sequence length used for training is longer. The loss function is most sensitive to the 10th output in the sequence, as this has the largest scale. This causes some of the model instances to just learn the mean, which becomes passable for very long sequences, which is why the success-rate increases for longer sequences. However, this is not a valid solution. A well-behavior model should be successful independent of the sequence length.

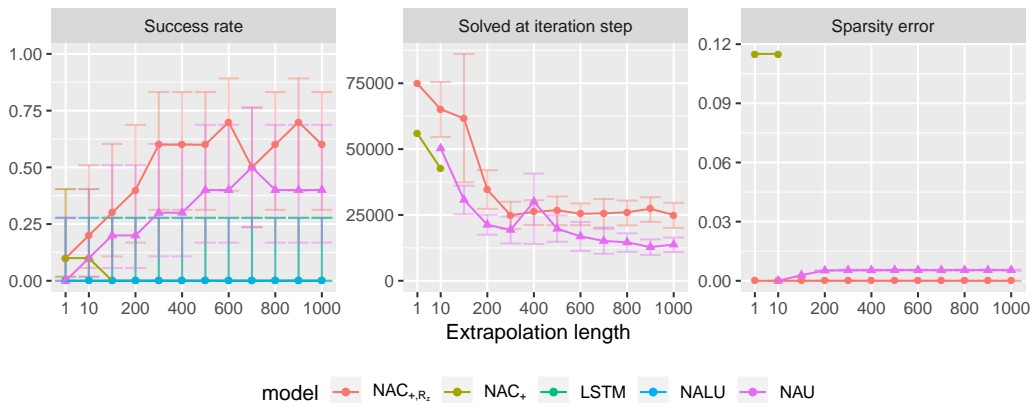

Figure 14: Same as figure 13, but where the NAU model do not use the $R_z$ regularizer.

## D.4 SEQUENTIAL MULTIPLICATION WITHOUT THE $R_z$ REGULARIZER

As an ablation study of the $R_z$ regularizer figure 15 shows the NMU and $NAC_{\bullet,NMU}$ models without the $R_z$ regularizer. The success-rate is somewhat similar to figure 4. However, as seen in the "sparsity error" plot, the solution is quite different.

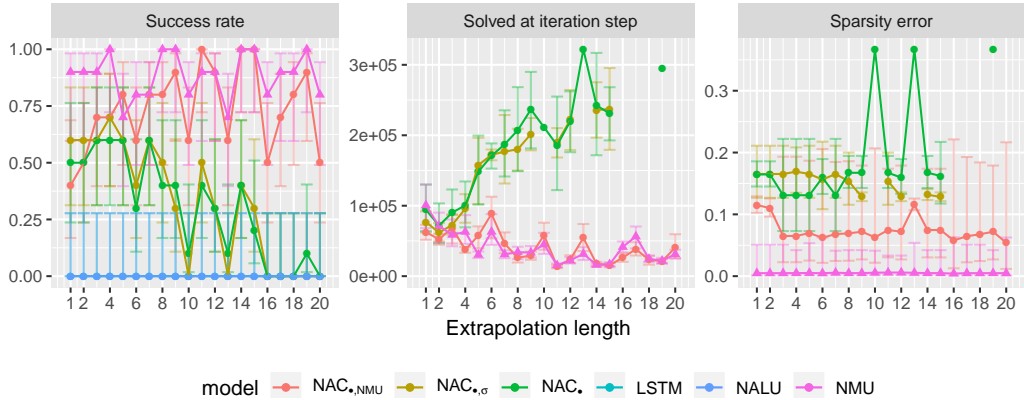

Figure 15: Shows the ability of each model to learn the arithmetic operation of addition and back-propagate through the arithmetic layer in order to learn an image-to-scalar value for MNIST digits. The model is tested by extrapolating to larger sequence lengths than what it has been trained on. The NMU and $NAC_{\bullet,NMU}$ models do not use the $R_z$ regularizer.

