# OpenReview forum: "Neural Arithmetic Units"
_ICLR.cc/2020/Conference — Accept (Spotlight)_

### Official Review · AnonReviewer2 · 2019-10-23
**Official Blind Review #2**

**Rating:** 6

**Review:**

The authors propose the Neural Multiplication Unit (NMU), which can learn to solve a family of arithmetic operations using -, + and * atomic operations over real numbers from examples. They show that a combination of careful initialization, regularization and structural choices allows their model to learn more reliably and efficiently than the previously published Neural Arithmetic Logic Unit.

The NALU consists of two additive sub-units in the real and log-space respectively, which allows it to handle both additions/subtractions and multiplications/divisions, and combines them with a gating mechanism. The NMU on the other hand simply learns a product of affine transformations of the input. This choice prevents the model from learning divisions, which the authors argue made learning unstable for the NALU case, but allows for an a priori better initialization and dispenses with the gating which is empirically hard to learn. The departures from the NALU architecture are well justified and lead to significant improvements for the considered applications, especially as far as extrapolation to inputs outside of the training domain.

The paper is mostly well written (one notable exception: the form of the loss function is not given explicitly anywhere in the paper) and well executed, but the scope of the work is somewhat limited, and the authors fail to properly motivate the application or put it in a wider context.

First, divisions being difficult to handle does not constitute a sufficient justification for choosing to exclude them: the authors should at the very least propose a plausible way forward for future work. More generally, the proposed unit needs to be exposed to at least 10K examples to learn a single expression with fewer than 10 inputs (and the success rate already drops to under 65% for 10 inputs). What would be the use case for such a unit? Even the NMU is only proposed as a step on the way to a more modular, general-purpose, or efficient architecture, its value is difficult to gauge without some idea of what that would look like.



**Experience Assessment:**

I have read many papers in this area.

**Review Assessment: Checking Correctness Of Derivations And Theory:**

I assessed the sensibility of the derivations and theory.

**Review Assessment: Checking Correctness Of Experiments:**

I assessed the sensibility of the experiments.

**Review Assessment: Thoroughness In Paper Reading:**

I read the paper thoroughly.

---

> ### Author Response · Authors · 2019-11-08
> **Response to reviewer #2 - thank you for your review**
>
> Dear reviewer #2, we thank you for your review and in particular your feedback on our experimental section. As is often the case with foundational research, the applications are not always immediately clear. We belive that multiplication is useful, however as both NALU and NMU are very recent additions to the field of neural networks, the best applications have yet to emerge. We elaborate on what applications we think multiplication can be applied to below. We would appreciate further feedback and hope that we can employ some of your concerns to strengthen our experimental section.
>
> - divisions being difficult to handle does not constitute a sufficient justification for choosing to exclude them: the authors should at the very least propose a plausible way forward for future work.
>
> We understand your concerns, it has been challenging for us to write our results. To be honest we don’t believe that division actually works for NALU.
>
> That division doesn’t work is apparent when carefully inspecting Table 1 in the NALU paper. Here the results shows that division on interpolation doesn’t work, but it does work for extrapolation. Given the construction of NALU, it should be clear that if the model had truly found a correct solution, it should work for both interpolation and extrapolation. Unfortunately, due to the reporting of results in NALU [table 1] bad models can appear to be correctly converged as their comparison is based on a relative improvement over a random baseline model (details in our reviewer #4 response). This is mentioned in Appendix C.7.1.
>
> This motivated us to change the evaluation criteria. We have published an in-depth explanation of these issues as well as a reproduction-study of NALU (shows the same results) in the SEDL workshop at NeurIPS 2019. We have shared this reproduction-study (which includes a table showing that division doesn’t work) with the authors of NALU, where the first author Andrew Trask publicly responded “Great work! We can’t improve without good benchmarks.”. We have made an anonymized version of the paper available here: https://www.dropbox.com/s/e03kd4x9j0l7b5b/Measuring_Arithmetic_Extrapolation_Performance.pdf?dl=0 (please respect the double-blinded process, as the non-anonymous is on arXiv).
>
> - More generally, the proposed unit needs to be exposed to at least 10K examples to learn a single expression with fewer than 10 inputs (and the success rate already drops to under 65% for 10 inputs).
>
> The complexity of the problem (hidden size, Figure 3) is indeed illusive. A good way to understand the complexity of these problems is to linearize them, such that they can be solved with a linear regression. Take for example the simple case from section 1.1, (x_1 + x_2)  * (x_1 + x_2 + x_3 + x_4). An alternative way to learn this problem would be to expand the input vector to include all possible combinations. In this case it would be [x1, x2, x3, x4, x1*x1, x1*x2, x1*x3, x1*x4, x2*x2, x2*x3, x2*x4, x3*x3, x3*x4, x4*x4]. A linear regression could then learn to sum the correct values. For 10 hidden size in Figure 3, this is much more complex as the input size is 100 and we allow up to 10 subsets to be multiplied. To compute the linearized size use Sum(choose(100 + i - 1,i), i=1..10) = 46897636623980, which is a huge input size for a linear regression. We hope that this gives some intuition as to why it is such a challenging problem.
>
>  - What would be the use case for such a unit? Even the NMU is only proposed as a step on the way to a more modular, general-purpose, or efficient architecture, its value is difficult to gauge without some idea of what that would look like.
>
> When building a basic component, SOTA results on a commonly known benchmark always help the story! However, we believe that the subject of arithmetic extrapolation is still in its infancy and might need more time before it is used ubiquitous. As explicit arithmetic and logical constructs are rarely present in the type of datasets commonly used for evaluating machine learning models (e.g. NLP), we would need to work with individuals that knowledge and access to such data, in order to better understand how we should integrate the NMU with common deep learning contraptions such as the LSTM. In particular, we think that unknown differential equations, or physical models, might be a good application of the NMU. However, in this work, our main concern has been to uncover and overcome some of the theoretical concerns of the NALU and build a component that can work with high number of hidden states, which is necessary in deep neural networks.

---

### Official Review · AnonReviewer3 · 2019-10-24
**Official Blind Review #3**

**Rating:** 6

**Review:**

The authors extend the work of Trask et al 2018 by developing alternatives to the Neural Accumulator (NAC) and Neural Arithmetic Logic Unit (NALU) which they dub the Neural Addition Unit (NAU) and Neural Multiplication Unit (NMU), which are neural modules capable of performing addition/subtraction and multiplication, respectively. The authors show that their proposed modules are capable of performing arithmetic tasks with higher accuracy, faster convergence, and more theoretically well-grounded foundations.

The new modules modules are relatively novel, and significantly outperform their closest architectural relatives, both in accuracy and convergence time. The authors also go to significant lengths to demonstrate that the parameters in these modules can be initialized and learned in a more theoretically well-grounded manner than their NAC/NALU counterparts. For these reasons I believe this paper should be accepted.

General advice/feedback:
- should provide an explanation of the row in Table 2 showing that a simple linear transformation is able to achieve accuracy and convergence times comparable to those of the NAU
- should provide an explanation of the universal 0% success rate on the U[1.1,1.2] sampling interval in Figure 3
- inconsistent captioning in Figure 2c, missing "NAC• with"
- should clarify in Section 4.1 that the "arithmetic dataset" task involves summing only *contiguous* vector entries; this is implied by the summation notation, and made explicit in Appendix Section C, but not specified in Section 4.1
- it is unclear what experiments you performed to obtain Figure 3, and the additional explanation in Appendix Section C.4 regarding interpolation/extrapolation intervals only adds to the confusion; please clarify the explanation of Figure 3, or else move it to the Appendix
- the ordering of some of the sections/figures is confusing and nonstandard: Section 1.1 presents results before explaining what exactly is being measured, Figure 1 shows an illustration of an NMU 2 pages before it is defined, Section 3 could be merged with the Introduction

Grammatical/Typesetting errors:
- "an theoretical" : bottom of pg 2
- "also found empirically in (see Trask et al. (2018)" : top of pg 4
- "seamlessly randomly" : middle of pg 5
- "We choice" : middle of pg 6
- inconsistent typesetting of "NAC" : bottom of pg 6
- "hindre" : middle of pg 8
- "to backpropergation" : bottom of pg 8
- "=≈" : top of pg 17
- "mathcalR" : bottom of pg 23
- "interrest" : bottom of pg 24
- "employees" : bottom of pg 24
- "models, to" : bottom of pg 24
- "difference, is" : bottom of pg 24
- "consider them" : bottom of pg 24
- "model, is" : top of pg 25
- "task, is" : top of pg 25
- "still struggle" : top of pg 25
- "seam" : top of pg 27
- "inline" : top of pg 27
- inconsistent typesetting of "NAC" : top of pg 27


**Experience Assessment:**

I have read many papers in this area.

**Review Assessment: Checking Correctness Of Derivations And Theory:**

I assessed the sensibility of the derivations and theory.

**Review Assessment: Checking Correctness Of Experiments:**

I assessed the sensibility of the experiments.

**Review Assessment: Thoroughness In Paper Reading:**

I read the paper thoroughly.

---

> ### Author Response · Authors · 2019-11-08
> **Response to reviewer #3, thanks for your thorough review**
>
> Dear reviewer #3, thank you for your kind words and thorough review, it is most appreciated.
>
> - should provide an explanation of the row in Table 2 showing that a simple linear transformation is able to achieve accuracy and convergence times comparable to those of the NAU
>
> Thanks, we have added “For addition NAU is comparable to a linear transformation in success-rate and convergence speed, but is more sparse.”
>
> - inconsistent captioning in Figure 2c, missing "NAC• with"
>
> Thanks, this has been corrected.
>
> - should clarify in Section 4.1 that the "arithmetic dataset" task involves summing only *contiguous* vector entries; this is implied by the summation notation, and made explicit in Appendix Section C, but not specified in Section 4.1
>
> Thanks, this has been added. Although note that as the first layers is a linear layer, it is invariant to the order of the elements.
>
> - it is unclear what experiments you performed to obtain Figure 3, and the additional explanation in Appendix Section C.4 regarding interpolation/extrapolation intervals only adds to the confusion; please clarify the explanation of Figure 3, or else move it to the Appendix
>
> Thanks, the extrapolation ranges need to be changed to not overlap with the interpolation range also reflect the scale of the interpolation range. We have made this more explicit in both the results section and appendix. All other parameters are unchanged. Let us know if this is still confusing.
>
> - should provide an explanation of the universal 0% success rate on the U[1.1,1.2] sampling interval in Figure 3
>
> Thanks for pointing out the add behaviour of 0% success rate on U[1.1, 1.2]. We simply do not know why that is the case. However, as it was a part of our original testing for interpolation-range sensitivity we have kept it in the plot. We added the following comment in our result section discussing the figure: Interestingly, none of the models can learn the $\mathrm{U}[-1.1,1.2]$, suggesting that certain input distributions might be troublesome to learn.”
>
> - the ordering of some of the sections/figures is confusing and nonstandard: Section 1.1 presents results before explaining what exactly is being measured, Figure 1 shows an illustration of an NMU 2 pages before it is defined, Section 3 could be merged with the Introduction
>
> This is true, we use 1.1 to provide the reader with an example on what we are trying to solve and to highlight the challenges with NALU which motivates why we are looking at multiplication. We focus mainly on what the data input is and what the optimal solution is. We believe this problem introduction is important to give the reader a softer introduction before we begin the more formal mathematical descriptions in our method section. Keep in mind that not everybody is as familiar with arithmetic extrapolation as compared to other more typical subjects.
>
> We do acknowledge that this is not the usual way of presenting a problem. If you believe that this negatively impacts the reading experience and your review, we would gladly either change, replace or completely remove this sub-section.
>
> - Grammatical/Typesetting errors:
>
> Thanks, we truly appreciate your thoroughness.

---

### Official Review · AnonReviewer1 · 2019-10-29
**Official Blind Review #1**

**Rating:** 3

**Review:**

This paper aims to address several issues shown in the Neural Arithmetic Logic Unit, including the unstability in training, speed of convergence and interpretability. The paper proposes a simiplification of the paramter matrix  to produce a better gradient signal, a sparsity regularizer to create a better inductive bias, and a multiplication unit that can be optimally initialized and supports both negative and small numbers.

As a non-expert in this area, I find the paper interesting but a little bit incremental. The improvement for the NAC-addition is based on the analysis of the gradients in NALU. The modification is simple. The proposed neural addition unit uses a linear weight design and an additional sparsity regularizer. However, I will need more intuitions to see whether this is a good design or not. From the experimental perspective, it seems to work well.
Compared to NAU-multiplication, the Neural Multiplication Unit can represent input of both negative and positive values, although it does not support multiplication by design. The experiments show some gain from the proposed NAU and NMU.

I think the paper can be made more self-contained. I have to go through the NALU paper over and over again to understand some claims of this paper. Overall, I think the paper makes an useful improvement over the NALU, but the intuition and motivation behind is not very clear to me. I think the authors can strengthen the paper by giving more intuitive examples to validate the superiority of the NAU and NMU.

**Experience Assessment:**

I do not know much about this area.

**Review Assessment: Checking Correctness Of Derivations And Theory:**

I assessed the sensibility of the derivations and theory.

**Review Assessment: Checking Correctness Of Experiments:**

I assessed the sensibility of the experiments.

**Review Assessment: Thoroughness In Paper Reading:**

I read the paper at least twice and used my best judgement in assessing the paper.

---

> ### Author Response · Authors · 2019-11-08
> **Response to reviewer #1 - we appreciate your feedback**
>
> Dear Reviewer #1, thank you for your valuable comments and insight. Writing this paper was not easy as we had to juggle theoretical findings, making a new evaluation criterion, finding appropriate tasks, a reproduction study of the NALU, and providing evidence of our new methods. We are grateful for your feedback and would love to collaborate with you on how to best present our findings. Below we have taken snippets of your comments and either modified our submission or provided further questions to get clarification. We appreciate your time and effort.
>
> - The proposed neural addition unit uses a linear weight design and an additional sparsity regularizer. However, I will need more intuitions to see whether this is a good design or not.
>
> A linear function is always easier to fit than a non-linear function (for example NALU’s tanh(x)sigmoid(x) weights). We try to elaborate upon this in section 2.2, where we attempt to provide a theoretical analysis of the gradients from the tanh(x)sigmoid(x) weights construct provided by the original NALU paper. Our findings suggest that optimal initialization causes the gradients to be zero. The sparsity regularizer bias the weights to {-1,0,1} which tanh(x)sigmoid(x) unfortunately does not.
>
> A sparse solution is often an intrinsic property in the problem domain of arithmetic extrapolation. For example, all the experiments in the NALU paper have this property. Furthermore, even when it is not an intrinsic property, say for example we need to learn 1.5*x1*x2, the arithmetic rules of addition and multiplication mean that these constants can always be learned by another more traditional layer. In this example, a linear transform can learn to multiply x1 by 1.5, or simply add a constant as one of its hidden outputs. Therefore, the bias restricts our optimization space allowing us to find exact solutions but does in combination with traditional layers not restrict what solutions can be found.
>
> We have added the following to section 2.2:  “This bias is desired as it restricts the solution space to exact addition, and in section 2.5 also exact multiplication, which is an intrinsic property of an underlying arithmetic function. However, it does not necessarily restrict the output space as a plain linear transformation will always be able to scale values accordingly. The bias also adds interpretability which is important for being confident in a model’s ability to extrapolate.”
>
> - I have to go through the NALU paper over and over again to understand some claims of this paper”, “I think the paper can be made more self-contained”
>
> In the tradeoff between describing the NALU paper and focussing on our own contributions we have chosen to restrict the description of NALU to section 2.1. We are happy to update the paper, so please suggest changes that would help reading.
>
> - Overall, I think the paper makes an useful improvement over the NALU, but the intuition and motivation behind is not very clear to me.
> I think the authors can strengthen the paper by giving more intuitive examples to validate the superiority of the NAU and NMU.
>
> Including division in NALU means that there is a singularity in the optimization space. As you can see in Figure 2, this leads to a dangerous optimization space where unwanted minimas are close to singularities. You will also see that when division is removed the NAC performs significantly better for a hidden size of 2.
>
> Initialization is not very important if the hidden size is 2. However, when the hidden size becomes larger optimal initialization is often important. We understand that optimal initialization is not an intuitive subject, but we hope that it is clear that NAC_mul cannot be optimally initialized and that NMU can, and that this is what gives much better performance for a larger hidden size.
>
> We hope that this clarifies things. As we do believe this is already described we it would be very helpful if you could pinpoint which paragraphs lack intuition.

---

### Official Review · AnonReviewer4 · 2019-11-03
**Official Blind Review #4**

**Rating:** 8

**Review:**

DISCLAIMER: I reviewed a previous version of this paper at another venue.

This paper introduces Neural Addition Units (NAU) and Neural Multiplication Units (NMU), essentially redeveloped models of the Neural Arithmetic Logic Unit (NALU). The paper presents a strong case that the new models outperform NALUs in a few metrics: rate of convergence, learning speed, parameter number and model sparsity. The performance of NAU is better than NAC/NALU, as is the performance of NMU with a caveat that the presented NMU here cannot deal with division, though it can deal with negative numbers (as opposed to NALU).

What this paper excels at is a thorough theoretical and practical analysis of NALU’s issues and how the authors design the two new models to overcome these issues. The presented issues of NALU are numerous, including unstable optimization space, expectations of gradients converging to zero, the inability of NALUs gating to work as well as intended and its issues with division, and finally, the intended values of -1, 0, 1 in NALU do not get as close to these values as intended.

The paper is easy to read, modulo a number of typos and admittedly some weirdly written sentences (see typos and minor issues later) and I would definitely recommend another iteration over the text to improve the issues with it as well as the style of writing. I am quite fond of the analysis and the informed design of the two new models, as well as the simplicity of the final models which are fairly close to the original models but have been shown both theoretically and practically that they work.
It is great to see that the paper improved since my last review and stands stronger on its results, but there are still a few issues with it that make me hesitant to fully accept the paper:
- The conclusion of the paper is biased towards the introduced models, but it should clearly define the limitations of these models wrt NALU/NAC
- The performance of NALu on multiplication is in stark contrast to the results in the original paper (Table 1). This should be commented in the paper why that is, as the original model presents no issues of NALU with multiplication, whereas this paper essentially says that they haven’t gotten a single model (out of 100 of them) to do multiplication.
- Could you explicitly comment on the paper why is the parameter sparsity such a sought-after quality of these models?
- You ‘assume an approximate discrete solution with parameters close to {1-, 0, 1} is important’. What do you have to back this assumption? Would it be possible to learn the arithmetic operations (and generalize) even with parameters different than those?
- Why did you introduce the product of the sequential MNIST experiment but did not presents results on the original sum / counting of digits? The change makes it hard to compare with the results in the original paper, and you do not present the reason why. This also makes me ask why didn't you compare to NALU on more tasks presented in the paper?

To conclude, this paper presents a well-done experimental and theoretical analysis of the issues of NALU and ways to fix it. Though the models presented outperform NALU, they still come with their own issues, namely they do not support division, and (admittedly, well corroborated with analysis) are not joined in a single, NALU-like model, that can learn multiple arithmetic operations. The paper does a great analysis of the models’ issues, with an experimental setup that highlights these issues, however, it does that on only one task from the original paper, and a(n insufficiently justified) modification of another one (multiplication of MNIST digits)---it does not extensively test these models on the same experimental setup as the original paper does.

Typos and smaller issues:
- Throughout the text you say that NMU supports large hidden input sizes? Why hidden??
- Figure 4 is identical to figure in D.2
- Repetition that E[z] = 0 is a desired property in 2.2, 2.3, 2.4
- In Related work, binary representation -> one-hot representation
- Found empirically in () - remove parentheses and see
- increasing the hidden size -> hidden vector size?
- NAU and NMU converges/learns/doesobtains -> converge/learn/do/obtain
- hard learn -> hard to learn ?
- NAU and NMU ...and improves -> improve
- Table 1 show -> shows
- Caption Table 1: Shows the - quite unusual caption (treating Table 1 as part of the sentence), would suggest to rephrase (e.g. Comparison/results of … on the … task). Similarly for Table 2...and Figure 3
- experiemnts -> experiments
- To analyse the impact of each improvements….. - this sentence is missing a chunk of it, or To should be replaced by We
- Allows NAC_+ to be -> remove be
- can be express as -> expressed
- The Neural Arithmetic Expression Calculator () propose learning - one might read this as the model proposes, not the authors / paper / citation propose…(also combine or combines in the next line)
- That the NMU models works -> model works? models work?
- We choice the -> we choose the
- hindre -> hinder
- C.5 seperate -> separate
- There’s a number of typos in the appendix
- convergence the first -> convergence of the first?
- Where the purpose is to fit an unknown function -> I think a more appropriate statement would hint at an often overparameterization in practice done when fitting a(n unknown) function


**Experience Assessment:**

I have read many papers in this area.

**Review Assessment: Checking Correctness Of Derivations And Theory:**

I assessed the sensibility of the derivations and theory.

**Review Assessment: Checking Correctness Of Experiments:**

I assessed the sensibility of the experiments.

**Review Assessment: Thoroughness In Paper Reading:**

I read the paper thoroughly.

---

> ### Author Response · Authors · 2019-11-08
> **Response to reviewer #4 - regarding other NALU experiments**
>
>
>
> - Why did you introduce the product of the sequential MNIST experiment but did not presents results on the original sum / counting of digits? ...
>
> The major argument of using the NALU is extrapolation, multiplication, and plug-in integration with neural networks. 4.1 tests extrapolation and 4.2 can, without major modifications, test multiplication in integration with a larger network (CNN).
>
> While we do propose the NAU, the main focus of our paper is the NMU. We do not think investigating the NAC+ is particularly interesting as it works. As a result our experiments focuses on multiplication.
>
> Below we have elaborated on the tasks of the NALU paper and why we believe that they fit/do not fit the purpose of: extrapolation, plug-in integration with neural networks, and multiplication.
>
> 4.1 Simple Function Learning Tasks
> -Extrapolation: Numeric extrapolation can be achieved by increasing the input/output range
> -Integration with neural networks: By increasing the hidden-size we can assess the theoretical modeling capacity of these units.
> -Multiplication: is explicitly tested.
>
> In the original paper dataset hyperparameters are not reported, which is why we choose to extensively test various combinations.
>
> 4.2 MNIST Counting and Arithmetic Tasks
> -Extrapolation: This experiment does not test value-extrapolation (the primary goal of NALU), as the network needs to see all digits. The sequential extrapolation is a different type of extrapolation that relates more to getting precise sparse values, as minor errors will accumulate exponentially.
> -Integration with neural networks: It does not integrate with a neural network, but by placing the arithmetic component after a CNN we can the arithmetic units capabilities and how well gradient signal travels through the arithmetic units.
> -Multiplication: While it is called “Arithmetic tasks”, they only test for addition.
>
> We choose to extend this to multiplication as it is the focal point of our paper.  We will run the tasks with our NAU for comparison, which we will report in the appendix when the results are ready.
> To further elaborate, we added the following description to our introduction “We propose the MNIST multiplication variant as we want to test the NMU's and 's ability to learn from real, noisy data where the numeric input has to be learned from features.“
>
> 4.3 Language to Number Translation Tasks
> -Extrapolation: This task does not pose any extrapolation requirements, as the test set consists of numbers in the training range.
> -Integration with neural networks: The arithmetic layer could be placed in the recurrent connection and use the operands to choose between gating. However, when contacting the main author about their architecture we find that they do not use their arithmetic components in the recurrent layer. Instead they use it to modify the final output, which means that all arithmetic modeling is performed by the LSTM (here is an anonymous link to the architecture that the main author has agreed on in our email correspondence: https://ibb.co/x7J1FZg).
> -Multiplication: Multiplication is not required. This may be counter-intuitive, but the network does not need to learn multiplication to produce 7*100+2 = 702, as the network always multiplies by 100 and therefore it can be learned using a linear layer.
>
> Given this does not test for extrapolation or multiplication, we have chosen not to include this task.
>
> 4.4 Program Evaluation
> -Extrapolation: is tested
> -Integration with neural networks: also tested
> -Multiplication: no multiplication is tested. They describe the experiment as “The first consists of simply adding two large integers, and the latter involves evaluating programs containing several operations (if statements, +, −).“
>
> Given this does not test multiplication, we have chosen to not include this experiment.
>
> 4.5 Learning to Track Time in a Grid-World Environment
> -Extrapolation: This task requires extrapolation when testing on numeric “waiting” ranges above the training range.
> -Integration with neural networks: As detailed by the authors, arithmetic components needs to be integrated into the architecture.
> -Multiplication: This task concerns counting, counting can be solved by an LSTM as shown in formal language work (https://arxiv.org/abs/1805.04908).
>
> Because this task only tests counting, we do not think it is interesting. Furthermore, it is difficult to implement and the authors provide no code for this. We have asked the authors for the code, but they were not able to help us.
>
> 4.6 MNIST Parity Prediction Task & Ablation Study
> -Extrapolation: As mentioned explicitly in the NALU paper, this task is designed for interpolation.
> -Integration with neural networks: The arithmetic unit integrates with a larger network as described in Seguí et al.
> -Multiplication: This task is designed for addition and not multiplication.
>
> Because of the lack of extrapolation and multiplication we have chosen not to include this task.

---

> > ### Author Response · Authors · 2019-11-13
> > **Added sequential addition of MNIST digests results**
> >
> > Dear reviewer. We have now also added results for the sequential addition of MNIST digests under Appendix D. We hope this will satisfy your concerns.

---

> ### Author Response · Authors · 2019-11-08
> **Response to reviewer #4 - addressing specific comments**
>
> - The conclusion of the paper is biased towards the introduced models, but it should clearly define the limitations of these models wrt NALU/NAC
>
> Thanks for pointing this, we have updated our conclusion with: “Our study shows that gating behaves close to random for both NALU and a gated NMU/NAU variant. However, when the gate correctly selects multiplication our NMU converges much more consistently.”
>
> Furthermore, as part of our gating-analysis (Appendix C.5) we have added a unit that gates between NMU and NAU similarly to NALU. The results show that a sigmoid gating-mechanism between the NMU and NAU has similar gating-converges results (close to random). We hope that this adds clarity to what is the limitations of NAU and NMU and what are the limitations of sigmoid gating.
>
> - The performance of NALU on multiplication is in stark contrast to the results in the original paper (Table 1). This should be commented in the paper why that is, as the original model presents no issues of NALU with multiplication, whereas this paper essentially says that they haven’t gotten a single model (out of 100 of them) to do multiplication.
>
> Originally we wanted to use the NALU for building NLP applications. However, as we investigated the unit we found that it was difficult to train and very fragile, which the main author agreed on over email correspondence. Deep diving into the result section of the NALU we found that their results, [Table 1], are easily misinterpreted. For example, the table shows that division on interpolation doesn’t work, but it does work for extrapolation. Given the construction of NALU, it should be clear that if the model had truly found a correct solution that should work for both interpolation and extrapolation. However, due to the reporting of results in NALU [table 1] bad models can appear to be correctly converged as their comparison is based on a relative improvement over a random baseline model. E.g. if the random baseline model has a loss of 1e10, a loss of 1e7 would be 0.001 using their reporting method. We choose to compare our results against a successful model instead, which is more interpretable and allows confidence intervals. Furthermore, our analysis of the convergence stability of a single NALU-gate (Appendix C.5) shows that convergence of gating is difficult and suggests cherry-picking of results.
>
> This is what motivated us to keep the experiment but change the evaluation criteria from a single-instance relative MSE to a success-criterium summarized over multiple seeds. We have published an in-depth explanation of these issues as well as a reproduction-study of NALU (shows the same results) in the SEDL workshop at NeurIPS 2019. We have shared this reproduction-study with the authors of NALU, where the main author Andrew Trask publicly responded “Great work! We can’t improve without good benchmarks.”. We have made an anonymized version of the paper available here: https://www.dropbox.com/s/e03kd4x9j0l7b5b/Measuring_Arithmetic_Extrapolation_Performance.pdf?dl=0
>
> - Could you explicitly comment on the paper why is the parameter sparsity such a sought-after quality of these models? ...
>
> The linear model on the subtraction problem is just the NAU without regularization and weight-clipping. Its success-rate is only 14%, while NAU and NAC_+ are 100%. This should justify that constraining to [-1, 1] is necessary. The sparsity regularizer itself is necessary as seen in the ablation study (Appendix C.3), although in that example both the regularizer and the clipping can be removed, with only a minor loss in success-rate and convergence speed.
>
> Generally speaking for Arithmetic Extrapolation, the fundamental assumption of the problem domain is that there is an exact solution of arithmetic operators that can be found. Because is always possible for a linear transformation to scale its output or add a learned constant output (linear bias), the bias towards {-1, 0, -1} does not restrict the output space, only the optimization space, when other traditional layers (that are based on linear transformations) are present.
>
> Furthermore having a sparse solution is much more interpretable which by manual inspection can add confidence in the model’s ability to extrapolate. We elaborated on this in section 2.2.
>
> - Throughout the text you say that NMU supports large hidden input sizes? Why hidden??
> Using the term “input size” could be misinterpreted as the network’s input size (the dimensions of the dataset).
>
> - Figure 4 is identical to figure in D.2
> We can understand the confusion! But by closely examining the values of the plot we find that the NMU has 100% success-rate in figure 4, but only 80% success-rate in figure D.2.
>
> - Repetition that E[z] = 0 is a desired property in 2.2, 2.3, 2.4
> This is correct, our goal has been to describe the issues independently of one another. Would you rather prefer that we reword this?

---

> ### Author Response · Authors · 2019-11-08
> **Response to reviewer #4 - general thanks and comments**
>
> Dear reviewer #4, thank you for your thorough review! We have tried our best to conform our paper to the feedback from our previous conference submission, in particular with elaborated results (MNIST), a better connection between theoretical findings and experimental design (testing increased hidden size), and a more fair comparison by excluding division from NAC_mul. Your comments are most useful, and we have updated our paper and responded to your questions using snippets of your review below. We are looking forward to a great discussion and hope to solve all the concerns you might have.

---

> > ### Comment · AnonReviewer4 · 2019-11-14
> > **Response + update**
> >
> > Thank you very much for the very detailed reply both to my review and all the other reviews. I’ve check the paper and read the entire correspondence.
> >
> > My comments are the following:
> > - Thank you for the C.5 experiments. Nice to see that the multiplication with gated NAU/NMU works better than with NALU, but still too bad that it seems that gating is a bad option for this model. This definitely adds to your claim that gating by itself is not a good choice here.
> > - Regarding the observation in the original NALU paper on division on interpolation and extrapolation, that is a good catch. Would you please comment on it in the appendix? In addition, please comment similarly (but short) on the findings of the workshop paper in the appendix.
> > - I would suggest (shortly) commenting the choice of MNIST multiplication vs counting and arithmetic tasks in the paper in the appendix to clarify it to future readers.
> >
> > Also, please do another grammar/style iteration over the paper as there are at least newly introduced issues, such as:
> > - for subtraction  a linear transformation can not -> cannot
> > - while NAU and NAC_+ solves -> solve
> > - as NMU and NAU uses -> use
> > - where an unknown function is often done by overparameterization -> overparameterization cannot fit functions, the model can…overparameterization in a property of the model wrt the function that needs to be fit!
> >
> >
> > My comments aside, I really think authors went out of their way to address our concerns in great detail. What I particularly like is that they did that at ICLR, i.e. on openreview so that this whole discussion will stay here out in the open for good as I see it as an important addition to the paper.
> >
> > I understand other reviewers’ concerns that the model presented in this paper is incremental, but I don’t see the strength of this paper to be just the model itself but the whole informed theoretical + experimental analysis leading to improvements of the model plus the open code which is there to stay (hopefully with the experimental setup too?), as opposed to the original paper. This paper nicely reads as a try to work with a recently presented models, the failure of the presented model and then a detailed process of analyzing and fixing the model and the benchmarks. The paper directly confronts the reproducibility issue with the original model, and improves drastically upon it. That is why I think this paper should definitely be accepted. I’m not sure whether the presented model will make a big change in the area, but the approach might influence and inspire other researchers to do more thorough analyses.
> >
> > Consequently, I’m increasing my score to accept.

---

> > > ### Author Response · Authors · 2019-11-14
> > > **Thanks for a productive dialog**
> > >
> > > Thanks for a productive dialog, both at ICLR and at the previous venue.
> > >
> > > - Regarding the observation in the original NALU paper on division on interpolation and extrapolation, that is a good catch. Would you please comment on it in the appendix?
> > >
> > > This is already mentioned in Appendix C.7.1. Let us know if you find it the explanation unclear.
> > >
> > > From section C.7.1.: >>Division does not work for any model, including the $\mathrm{NAC}_{\bullet}$ and NALU models. This may seem surprising but is actually in line with the results from the NALU paper (Trask et al., table 1) where there is a large error given the interpolation range. The extrapolation range has a smaller error, but this is an artifact of their evaluation method where they normalize with a random baseline. Since a random baseline will have a higher error for the extrapolation range, errors just appear to be smaller. A correct solution to division should have both a small interpolation and extrapolation error.<<
> > >
> > > - In addition, please comment similarly (but short) on the findings of the workshop paper in the appendix.
> > >
> > > We have added to appendix C: >>Our “arithmetic task” is identical to the “simple function task” in the NALU paper (Trask et al.,2018). However, as they do not describe their setup in details, we use the setup from Anonomous(2019), which provide Algorithm 3, an evaluation-criterion to if and when the model has converged, the sparsity error, as well as methods for computing confidence intervals for success-rate and the sparsity error.<<
> > >
> > > - I would suggest (shortly) commenting the choice of MNIST multiplication vs counting and arithmetic tasks in the paper in the appendix to clarify it to future readers.
> > >
> > > Thanks, we now elaborate on this in appendix D.1.: >>The sequential MNIST task takes the numerical value of a sequence of MNIST digits and applies a binary operation recursively. Such that $t_i = Op(t_{i-1}, z_t)$, where $z_t$ is the MNIST digit's numerical value. This is identical to the ``MNIST Counting and Arithmetic Tasks'' in Trask et al. [2018, section 4.2]. We present the addition variant to validate the NAU's ability to backpropagate, and we add an additional multiplication variant to validate the NMU's ability to backpropagate.<<
> > >
> > > We have also added references to Appendix D.2. in the results section.

---

### Author Response · Authors · 2019-11-08
**Deleted comments**

We apologize for the confusion with the deleted response messages. We had to delete them due to formatting errors related to the 5000 character limit. No text has been deleted. The new comments contain all the text from earlier.

---

### Author Response · Authors · 2019-11-13
**Summary of revision**

Dear reviewers, we appreciate your feedback. A lot of minor changes have been added in the last revisions and we hope we have addressed all of your concerns. To clarify what have been changed, we have made the following overview of major changes.

- Elaborated in the assumptions behind the sparsity bias.

Added in section 2.2.: This bias is desired as it restricts the solution space to exact addition, and in section \label{sec:method:nmu} also exact multiplication, which is an intrinsic property of an underlying arithmetic function. However, it does not necessarily restrict the output space as a plain linear transformation will always be able to scale values accordingly. The bias also adds interpretability which is important for being confident in a model’s ability to extrapolate.

- Elaborated on results for NAU.

Added in section 4.1.2.: For addition, NAU is comparable to a linear transformation in success-rate and convergence speed but is more sparse. However, for subtraction a linear transformation cannot consistently solve the task, while NAU and $\mathrm{NAC}_{+}$ solve it.

- Balanced conclusion regarding division and gating.

Added to section 5.: A natural next step would be to extend the NMU to support division and add gating between the NMU and NAU, to be comparable in theoretical features with NALU. However we find, both experimentally and theoretically, that learning the division is impractical, because of the singularity when dividing by zero, and that a sigmoid-gate that chooses between two functions with vastly different convergences properties, such as a multiplication unit and an addition unit, cannot be consistently learned.

- Added short summary of workshop publication for "Measureing Arithmetic Extrapolation Performance":

Added to section C.: Our “arithmetic task” is identical to the “simple function task” in the NALU paper (Trask et al.,2018). However, as they do not describe their setup in details, we use the setup from Anonomous(2019), which provide Algorithm 3, an evaluation-criterion to if and when the model has converged, the sparsity error, as well as methods for computing confidence intervals for success-rate and the sparsity error.

- Added Gated version of NAU/NMU, similar to NALU

Added to section C.5.: Furthermore, we also introduce a new gated unit that simply gates between our proposed NMU and NAU, using the same sigmoid gating-mechanism as in the NALU. This combination is done with separate weights, as NMU and NAU use different weight constrains and can therefore not be shared.

Added to section C.5.3. Which operation the gate converges to appears to be mostly random and independent of the task. These issues are caused by the sigmoid gating-mechanism and thus exists independent of the used sub-units.

Added to section C.5.: updated figure and results.

- Added results for sequential addition of MNIST digits

Previous section D.2. is now D.4.
New section D.2.: contains main results for "sequential addition of MNIST digits"
New section D.3.: contains an ablation study of $R_z$ the regularizer used in section D.2.

Added to section D.1.: The sequential MNIST task takes the numerical value of a sequence of MNIST digits and applies a binary operation recursively. Such that, where is the MNIST digit's numerical value. This is identical to the ``MNIST Counting and Arithmetic Tasks'' in Trask et al. [2018, section 4.2]. We present the addition variant to validate the NAU's ability to backpropagate, and we add an additional multiplication variant to validate the NMU's ability to backpropagate.

---

### Decision · Program_Chairs · 2019-12-19

**Decision:**

Accept (Spotlight)

**Comment:**

This paper extends work on NALUs, providing a pair of units which, in tandem, outperform NALUs. The reviewers were broadly in favour of the paper given the presentation and results. The one dissenting reviewer appears to not have had time to reconsider their score despite the main points of clarification being addressed in the revision. I am happy to err on the side of optimism here and assume they would be satisfied with the changes that came as an outcome of the discussion, and recommend acceptance.